# Single-cell RNA sequencing of the holothurian regenerating intestine reveals the pluripotency of the coelomic epithelium

**Joshua G Medina-Feliciano, Griselle Valentín-Tirado, Kiara Luna-Martínez, Alejandra Beltran-Rivera, Yamil Miranda-Negrón, José E Garcia-Arraras***

Department of Biology, University of Puerto Rico, San Juan, Puerto Rico

**\*For correspondence:**
jegarcia@hpcf.upr.edu

**Competing interest:** The authors declare that no competing interests exist.

## eLife Assessment

This article describes a resource detailing the econstitution of *Holothuria glaberrima* gut following self-evisceration in response to a potassium chloride injection, using scRNAseq and fluorescent RNA localization in situ. It provides some new findings about organ regeneration, as well as the origins of pluripotent cells, and places these findings in the context of regeneration across species. The article's schematic model and HCR images are a **valuable** foundation for future work. The authors provide **convincing** RNA localization images to validate their data and provide spatial context. These validation experiments are of good quality but remain challenging to connect to the complex spatial organization of complex tissues. This resource will be of interest to the field of regeneration, particularly in invertebrates, but also in comparative studies in other species, including evolutionary studies.

**Abstract** In holothurians, the regenerative process following evisceration involves the development of a 'rudiment' or 'anlage' at the injured end of the mesentery. This regenerating anlage plays a pivotal role in the formation of a new intestine. Despite its significance, our understanding of the molecular characteristics inherent to the constituent cells of this structure has remained limited. To address this gap, we employed state-of-the-art scRNA-seq and hybridization chain reaction fluorescent in situ hybridization analyses to discern the distinct cellular populations associated with the regeneration anlage. Through this approach, we successfully identified 13 distinct cell clusters. Among these, two clusters exhibit characteristics consistent with putative mesenchymal cells, while another four show features akin to coelomocyte cell populations. The remaining seven cell clusters collectively form a large group encompassing the coelomic epithelium of the regenerating anlage and mesentery. Within this large group of clusters, we recognized previously documented cell populations such as muscle precursors, neuroepithelial cells, and actively proliferating cells. Strikingly, our analysis provides data for identifying at least four other cellular populations that we define as the precursor cells of the growing anlage. Consequently, our findings strengthen the hypothesis that the coelomic epithelium of the anlage is a pluripotent tissue that gives rise to diverse cell types of the regenerating intestinal organ. Moreover, our results provide the initial view into the transcriptomic analysis of cell populations responsible for the amazing regenerative capabilities of echinoderms.

## Introduction

Animals exhibit a remarkable diversity in their responses to injury, ranging from basic wound healing to the complete regeneration of lost structures. At one end of the spectrum are species that heal wounds without regenerating the missing part, while at the other end are those capable of recreating

structures identical to the original. Despite these differences, all animals possess some level of regenerative ability. This capacity for regeneration can vary not only between different species but also within the same species, depending on the tissue or organ involved.

For decades, scientists have tried to understand these vast differences in regeneration capacities across the animal kingdom. For this they have focused on those species that show amazing regenerative abilities (*Tanaka and Reddien, 2011*), including coelenterates (hydra) (*Vogg et al., 2019*), flatworms (planaria) (*Reddien, 2018*), fish (zebrafish) (*Gemberling et al., 2013*), and amphibians (*Brockes and Kumar, 2002*; *Roy and Gatien, 2008*; *Beck et al., 2009*; *Joven et al., 2019*), among others. These studies have uncovered various mechanisms that regeneration-competent species exhibit to regenerate tissues, entire organs, body parts, and, in some cases, complete bodies. Key findings include the discovery of essential processes, such as the formation of a blastema – a mass of proliferating cells that plays a crucial role in regenerating the lost structure (*Min and Whited, 2023*; *Poleo et al., 2001*; *Santos-Ruiz et al., 2002*; *Seifert and Muneoka, 2018*; *Stocum, 2004*; *Zenjari et al., 1996*).

Among deuterostomes, echinoderms are considered prime exponents of regenerative capability (*Candia-Carnevali et al., 2024*). Within this group, holothurians, commonly known as sea cucumbers, exhibit an extraordinary form of regeneration. They can regenerate their internal organs following evisceration, a process in which they expel their viscera in response to stress or predation (*Byrne, 2023*). This extraordinary ability makes them a valuable model for studying regeneration in complex organisms. The regeneration of the digestive system in holothurians, in particular, has garnered significant interest (*Dolmatov, 2021*; *Mashanov et al., 2014*; *Medina-Feliciano and García-Arrarás, 2021*; *Quispe-Parra et al., 2021a*; *Su et al., 2022*). Upon evisceration, the holothurian intestine, which constitutes nearly their entire digestive tract, begins to regenerate from the mesentery, a supportive tissue layer where the original intestine was attached (*García-Arrarás, 1998*). *Figure 1 and Figure 1—figure supplement 1* provide a schematic view of the regeneration process and identify key structures for those not familiar with the holothurian model. A thickening at the injured end of the mesentery, known as an 'anlage' or 'rudiment', initiates this process (*García-Arrarás, 1998*). As will be discussed later, this structure is analogous to a blastema but differs in that the cell proliferation mainly occurs in the surrounding epithelium, rather than in the mesenchymal cells, as observed in classical blastemas (*Carlson, 2007*; *García-Arrarás, 1998*).

Histologically, the holothurian intestinal anlage forms from dedifferentiated cells within the mesentery, which revert to a more stem-cell-like state before proliferating and migrating to form a new intestinal structure (*Candelaria et al., 2006*; *Mashanov et al., 2005*). This dedifferentiation process is crucial for regeneration, involving a spatial and temporal gradient starting at the injury site and extending along the mesentery border (*García-Arrarás et al., 2011*). The new coelomic epithelial layer that forms around the anlage is distinct from the original mesothelium and shows significant morphological and molecular changes compared to the mesenteric tissue (*Candelaria et al., 2006*; *García-Arrarás, 1998*; *Mashanov et al., 2005*). Further examination of holothurian regeneration reveals that most cellular division occurs within the anlage's coelomic epithelium (*García-Arrarás, 1998*; *García-Arrarás et al., 2011*). These proliferating cells are hypothesized to differentiate into various cell types, including myocytes and neurons, as well as mesenchymal cells, through an epithelial to mesenchymal transition (EMT) (*García-Arrarás, 1998*; *García-Arrarás et al., 2011*). The gene expression profiles during the formation and growth of the anlage suggest extensive reprogramming, leading to a more plastic cell phenotype (*Ortiz-Pineda et al., 2009*; *Quispe-Parra et al., 2021b*; *Rojas-Cartagena et al., 2007*).

The process of intestinal regeneration in holothurians raises fundamental questions of regenerative phenomena, particularly concerning the identity, origin, and fate of progenitor cells, involved in the process (*Alvarado and Tsonis, 2006*; *Candia-Carnevali et al., 2024*). In this context, the role of the anlage, and specifically the mesothelium (also named celothelium) in echinoderms, deserved particular attention (*Candia-Carnevali et al., 2024*; *Smiley, 1994*). This tissue, mainly composed of coelomic epithelia and myocytes, exhibits significant morphological and gene expression changes that are associated with the dedifferentiation process. These dedifferentiated cells form the coelomic epithelium of the anlage and appear to be the principal source of cells for the new intestine. Despite these findings, little is known about the cell composition and dynamics of the anlage nor of its coelomic epithelium.

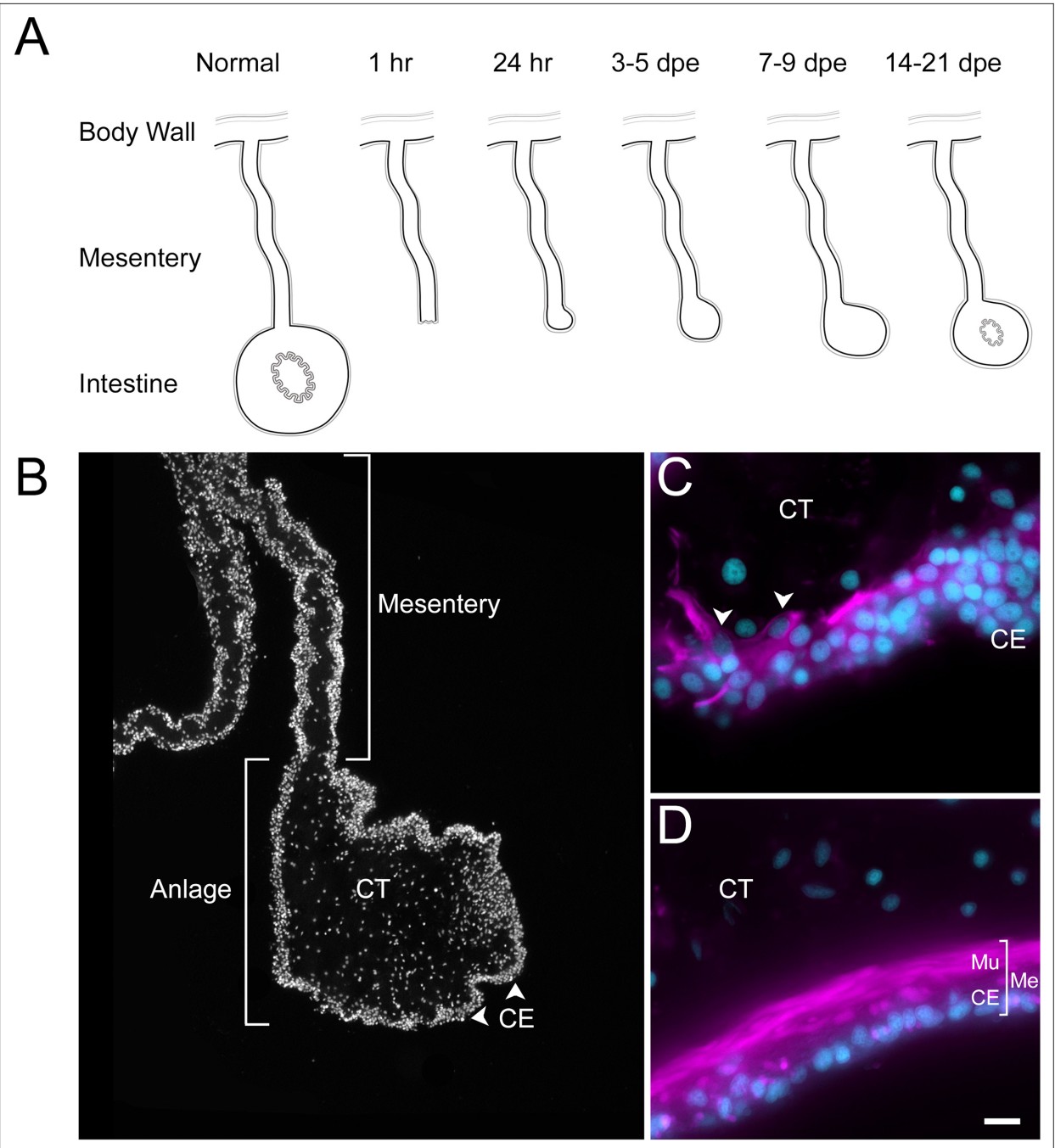

**Figure 1.** Intestinal regeneration in *H. glaberrima*. (**A**) Schematic view of normal and regenerating tissues as viewed in cross sections of normal and regenerating intestines. After evisceration (1 hr), the tip of the mesentery is torn but eventually is covered by CE (24 hr). A thickening of the mesentery tip forms the intestinal anlage, which grows in size for the next 2 weeks, initially (3–5 dpe) by cellular dedifferentiation and later by cellular proliferation (7–9 dpe). Eventually, the luminal epithelium is formed from migrating cells of the esophagus and cloacal ends of the digestive tract. (**B**) DAPI-stained section of 9-dpe anlage and mesentery showing the CE and CT layers. (**C, D**) Sections with fluorescently labeled phalloidin show the muscle (Mu) labeled in the (**C**) 9-dpe anlage and in the (**D**) normal intestine. The normal intestine is made of three distinct layers: mesothelium (Me) (that includes the CE and Mu), CT, and luminal epithelium (not shown). Me and CT are continuous throughout the mesentery and body wall. CE, coelomic epithelium; CT, connective tissue; Me, mesothelium; Mu, muscle; dpe, days post evisceration. Bar = 10 µm.

The online version of this article includes the following source data and figure supplement(s) for figure 1:

**Source data 1.** Antibody/markers used for immune- and cytochemical labeling of dissociated cell suspension.

**Figure supplement 1.** Overview of the intestinal regeneration process in the sea cucumber *H. glaberrima*.

*Figure 1 continued on next page*

*Figure 1 continued*

**Figure supplement 2.** Labeling of dissociated cell phenotypes with cell markers.

**Figure supplement 3.** Immunocytochemical labeling of dissociated cell phenotypes using three different antibodies against tubulin.

Single-cell RNA sequencing (scRNA-seq) offers tremendous promise to dissect the cellular contributions of the holothurian intestinal anlage and identify the specific cells involved in generating a new intestine. It is a powerful tool for dissecting cellular composition and dynamics that has been used in related species to explore regenerating or developing tissues, in mammals (*Ayyaz et al., 2019*; *Beumer and Clevers, 2021*; *Capdevila et al., 2021*; *Parigi et al., 2022*), axolotl and planaria (*Gerber et al., 2018*; *King et al., 2024*; *Leigh et al., 2018*; *Rodgers et al., 2020*), and echinoderm embryos (*Cocurullo et al., 2023*; *Paganos et al., 2022a*; *Paganos et al., 2021*; *Satoh et al., 2022*; *Tominaga et al., 2023*). These applications highlight the groundbreaking role of scRNA-seq in advancing our knowledge of the cellular mechanisms in both regenerative and developmental contexts.

In this study, we employ scRNA-seq to analyze the regenerating intestinal anlage of the sea cucumber *Holothuria glaberrima*, aiming to delineate its constituent cellular populations. We corroborate our findings using hybridization chain reaction fluorescent in situ hybridization (HCR-FISH) to verify the presence and location of specific cell types (*Choi et al., 2018*). The characterization of the cellular populations and their gene expressions serves to answer various questions such as: What are the cellular precursors? What is the role of the coelomic epithelium? What are the similarities and differences of the anlage and a classical blastema? This research not only advances our understanding of the unique regenerative capabilities of holothurians but also contributes to the broader field of regenerative biology, highlighting the diverse strategies employed by different organisms to restore lost tissues.

## Results

Previous work from our laboratory has shown that the rudiment or anlage that forms at the tip of the mesentery plays a pivotal role in the formation of the new intestine (*García-Arrarás et al., 2019*). This transient mass of cells is thought to give rise to most intestinal cell types, the sole exception being the luminal cells. Therefore, to maximize the characterization of the cells in the regenerative anlage, we chose to perform scRNA-seq in the tissues of 9-day post evisceration (dpe) regenerating animals (*García-Arrarás, 1998*; *García-Arrarás et al., 2011*). At this stage, a well-formed anlage consists of epithelial cells surrounding a large area of connective tissue populated with mesenchymal cells (*Figure 1A*). More importantly, different cell populations undergoing proliferation or differentiation can be found at this stage. Since some of the cellular processes during regeneration occur in a spatio-temporal gradient along the length of the mesentery, we separately surveyed the anlage and the mesentery tissues accordingly. Thus, for each animal, the anlage was separated from the mesentery, and both tissues were processed independently, for a total of four scRNA-seq runs.

### Cell heterogeneity: Immuno- and cytochemical analyses

The strength of the scRNA-seq data depends mainly on the dissociation and isolation of the cell populations from the dissected tissue. Since our focus was on the cells of the regenerating anlage, we devised a dissociation protocol that favored the isolation of the cells within this structure. To determine, at least partially, the cell types in our original dissociation, we performed immuno- and cytochemical analyses on the enzyme-dissociated cell suspension that was used for the scRNA-seq. The labeling obtained for each marker is shown in *Figure 1—figure supplement 2*.

We found that, in the dissociated anlage, 5% of the cells were labeled with a spherulocyte (immune cell) marker, 2–7% with phalloidin (a muscle marker), 40% with a mesenchymal marker (KL14-antibody), and a large number of cells (32–60%) with a mesothelial marker (MESO-antibody). Similar populations were found in the mesentery, although in this tissue, 5–10% of cells expressed the neuronal markers, heptapeptide GFSKLYFamide (*Blanco et al., 1995*), and RN1 (*Díaz-Balzac et al., 2007*). This analysis suggested that most of the cells originate from the regenerate coelomic epithelium.

Different populations were also observed with three different tubulin antibodies (*Figure 1—figure supplement 3*). In the 9-dpe anlage, anti-acetylated alpha tubulin labeled about 7% of the cells, anti-beta tubulin labeled about 70% of the cells, while an anti-alpha tubulin labeled about 80% of the cells.

Except for the acetylated alpha tubulin, which labeled about 1% of the cells dissociated from the mesentery versus 7% of those from the anlage, anti-alpha and beta labeling percentages were similar in both mesentery and anlage dissociated cells.

Finally, a cell population labeled with fluorescently labeled phalloidin accounted for about 7% of the cells in both mesentery and anlage. These cells, however, did not correspond to the elongated muscle cells of the mesentery. Instead, they were rounded cells with labeling found in the cytoplasm surrounding one side of the nuclei.

Two cell populations from the mesentery were absent or greatly underrepresented in the scRNA-seq. Firstly, the muscle cells of the mesentery, due to their non-dissociation by the protocol used, were not found in the dissociated cell suspension. Their elongated morphology would have further complicated their passage via cell separation system for sequencing. Secondly, the majority of neurons from the neuronal network associated with the mesentery (*Nieves-Ríos et al., 2020*) could not be dissociated. We did observe structures that resembled a tangled mass of cells immunoreactive to some of our neuronal markers. This suggests that the mesentery nervous component, being unable to be isolated as single cells, was not sequenced.

In summary, the immuno- and cytochemical results show that the dissociated cell populations sequenced correspond to cellular phenotypes that have been previously described within the regenerating anlage. The abundance of these cells in the sequenced samples corresponds to the ease of their dissociation by trypsin. Thus, dedifferentiated cells of the mesentery and anlage epithelium (which are loosely connected to each other) and those of the connective tissue are probably overrepresented compared to differentiated cell types.

## Cell populations defined by scRNA-seq

Analysis of scRNA-seq data resulted in a total of 3844 cells, with 2392 originating from the two anlage samples and 1452 from the two mesentery samples (*Figure 2—figure supplement 1*). Upon dataset integration and graph-based clustering, we identified 13 clusters, each thought to represent singular cell types or cell states in the regenerating intestine (*Figure 2A*, *Figure 2—figure supplement 2*). The percentage of cells that form each cluster differs from cluster to cluster, ranging from 21% (cluster 0) to 1% (cluster 12) of the total cells. Nonetheless, each cluster consists of cells from both the mesentery and anlage samples (*Figure 2B and C*). The number of clusters did not change dramatically under various parameters (resolution, dimensionality, and number of variable features), constantly around 12–15 clusters. Moreover, except for C3 and C4, the clusters were supported by the clustering significance analysis performed with scSHC (*Grabski et al., 2023*), a model-based hypothesis testing method for scRNA-seq that evaluates the probability of each individual cluster being unique (*Figure 2—figure supplement 2*). The uniqueness of C3 and C4 is suggested by additional analyses as will be addressed below.

Each identified cluster exhibited a distinctive gene expression profile relative to cells in other clusters (*Figure 2—figure supplement 3*). This shows the relative expression of the top gene for each cluster based on two factors: (1) difference in percentage of representation and (2) $\log_2$ fold-change ($\log_2$FC) against all other clusters. Interestingly, each cluster shows dramatic differential expression values (>2 $\log_2$FC) and differences in representation percentages over 50%, except for C0 through C3, with differences in representation around 30%.

Prior to characterizing each of the 13 cell clusters, we sought to understand what, in a broad view, appeared to be a segregation of ~90% of the cells into two distinct supra-clusters. One of them encompassing seven clusters (C0, C1, C3, C4, C5, C8, C9) that corresponded to 69.6% of all cells and the other encompassing two clusters (C2 and C7) that corresponded to 19.1% of cells. The remaining 11.3% of cells were distributed in four distinct isolated clusters (C6, C10, C11, and C12). As stated earlier, all clusters have representation from mesentery and anlage tissues (*Figure 2B and C*), thus excluding the possibility that the two supra-clusters represented mesentery versus anlage cells.

### Mesenchymal versus epithelial clusters

These two supra-clusters are of interest as they appear to represent the two main cell types found in the regenerating intestine: coelomic epithelial cells and mesenchymal cells (*Figure 2A*). Comparison between these two supra-clusters showed distinct expression profiles that allowed us to characterize their cell types (*Figure 3*, *Figure 2—figure supplement 3*). For instance, the supra-cluster composed

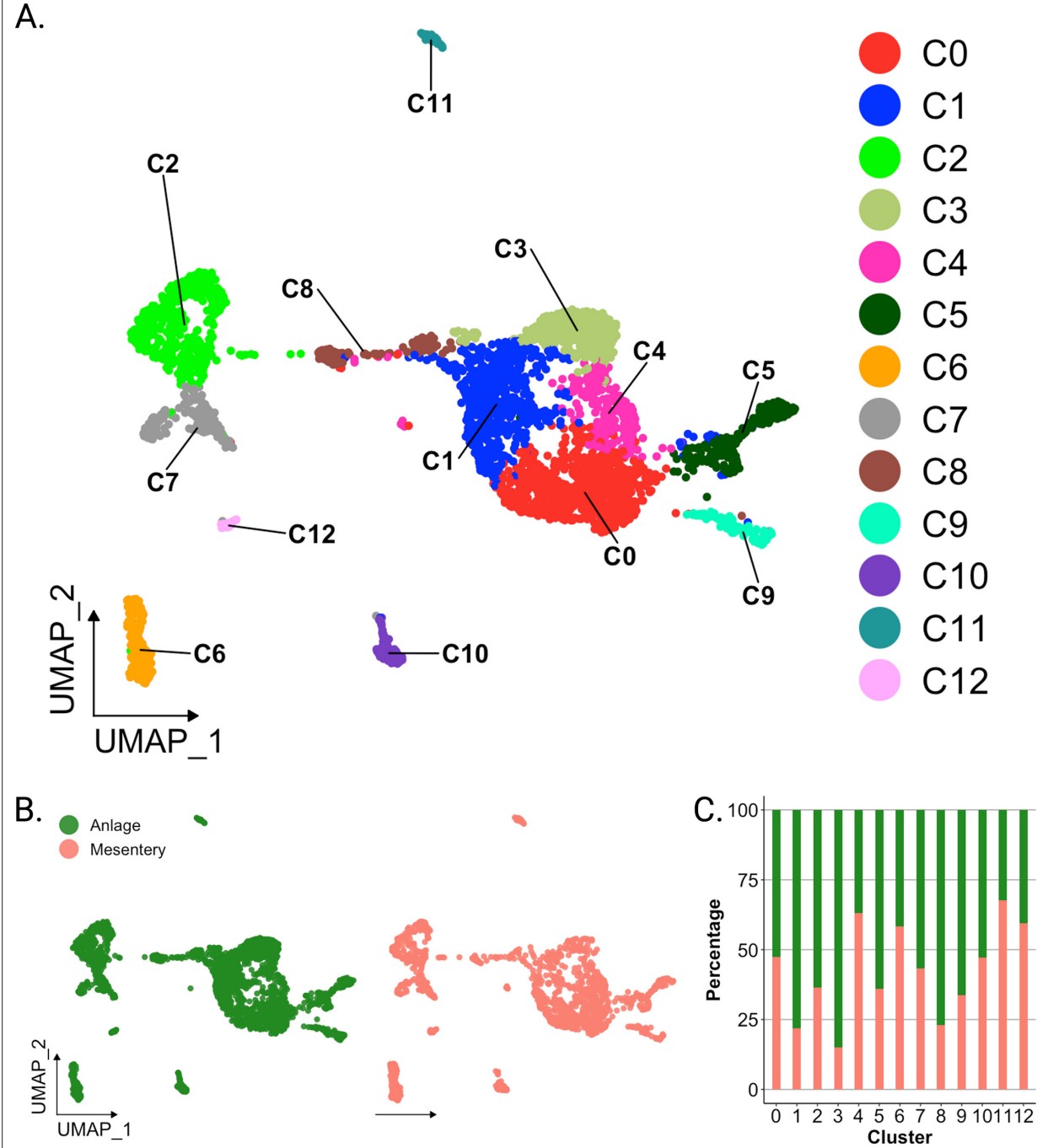

**Figure 2.** Overview of single-cell RNA sequencing of regenerating intestinal tissue of *H. glaberrima*. (**A**) UMAP plot of population identities determined through unsupervised clustering of 9-day regenerating mesentery and anlage tissues. (**B**) UMAP projections of cluster cells separated by tissue of origin. (**C**) Percentage of cells per cluster based on their tissue of origin.

The online version of this article includes the following source data and figure supplement(s) for figure 2:

**Source data 1.** General statistics of scRNA-seq data after mapping with Cell Ranger.

**Figure supplement 1.** Quality control assessment of *H. glaberrima* scRNA-seq data.

**Figure supplement 2.** UMAP of clusters after statistical assessment with scSHC.

**Figure supplement 3.** UMAP visualization of clusters highlighting the expression of their top genes.

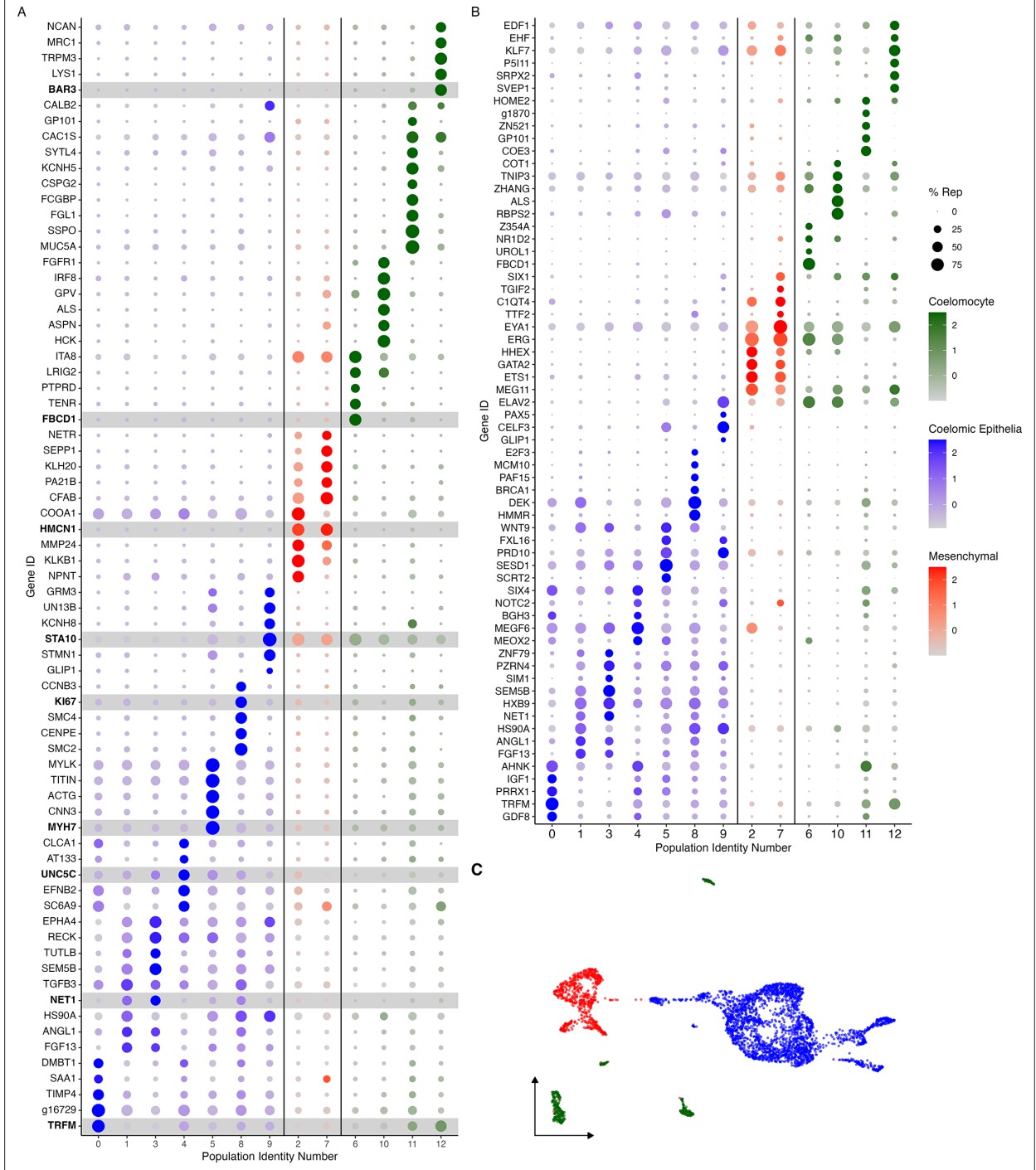

**Figure 3.** Cluster characterization by gene expression. (**A**) Top-expressed genes or (**B**) genes corresponding to transcription factors and intercellular signaling molecules are identified in the 13 cell clusters. Clusters are classified by their corresponding cell type, where blue corresponds to cells of the coelomic epithelium, red to those in the mesenchyme and green to coelomocytes. Color intensity shows the expression level of each gene in log$_2$fold-change (log$_2$FC) values. Dot size corresponds to percentage of representation of the gene in the respective cluster compared to all others. Gene identifiers starting with 'g' correspond to uncharacterized gene models of *H. glaberrima*. (**C**) UMAP plot of clusters colored by cell type.

The online version of this article includes the following figure supplement(s) for figure 3:

**Figure supplement 1.** Expression of marker genes previously documented in the sea cucumber.

**Figure supplement 2.** Enrichment of GO terms across cellular populations of the regenerative intestine of *H. glaberrima*.

**Figure supplement 3.** Expression of marker genes associated with cell types or state.

of C2 and C7, showed *ERG* (transcriptional regulator ETS-related gene) as the top expressed gene, an oncogene that is associated with mesenchymal cells in other echinoderms (**Meyer et al., 2023**; **Tominaga et al., 2023**). While this gene has a strong expression in C2 and C7, it also appears to be expressed in C6 and C10. However, it is important to highlight that *ERG* as a transcription factor has been shown to have roles in additional processes including inflammation and apoptosis.

The top marker gene in the other supra-cluster is *AHNK*, known as neuroblast differentiation-associated protein AHNAK. Reports have shown *AHNK* to have a role in calcium regulation, cellular migration, and carcinogenic transformation of colon epithelial cells (**Dumitru et al., 2013**). Furthermore, this gene is overexpressed in regenerating rat muscle compared to normal muscle (**Huang et al., 2007**). However, the localized expression of *WNT9* in the C1, C3, C5, and C8 of this supra-cluster more clearly favors its classification as a marker for coelomic epithelial cell types (**Figure 3B**, **Figure 3—figure supplement 1**). A previous study from our laboratory using in situ hybridization details the expression of *WNT9* during intestinal regeneration in *H. glaberrima,* where it was shown to be localized to the coelomic epithelium of the anlage and adjacent mesentery (**Mashanov et al., 2012**). In addition, correlating with what was shown by the in situ hybridization results, the population of cells that differentially expresses *WNT9* makes 15% of those in the mesentery. However, it is close to 30% of the cells in the anlage. Additional analyses, discussed below, further strengthen the coelomic epithelium identity of cells in this supra-cluster.

In summary, results show 13 individual cell clusters with distinct expression profiles in the regenerative intestinal tissue, with most of them corresponding to either mesenchymal or coelomic epithelia cells. The rest of the populations show top expressed genes that are immune-related, suggesting that these must be coelomocyte populations such as those that have been previously associated with both normal and regenerating intestinal tissues and that were also detected by immune and histochemistry in the cell samples used for the scRNA-seq (**Figure 1—figure supplement 2**; **García-Arrarás et al., 2006**; **Ramírez-Gómez et al., 2010**).

## Cluster identities

Rather than considering single genes, the uniqueness of each cluster can be assessed in terms of their transcriptomic profile (**Figure 3A**) and by the expression of transcription factors and intercellular signaling molecules (**Figure 3B**). As shown in **Figure 3A and B**, each cluster has a unique transcriptomic are clusters C0, C1, C3, and C4 of the epithelial layer populations, which share many top genes, albeit at different expression levels. Similarly, C8 and C9 have low expression of genes expressed by C0, C1, C3, and C4 but have expression of other genes that are not expressed by any other cluster. This is also true for the mesenchymal clusters, where these two clusters have overlap of some genes, that are not expressed by other clusters. Therefore, these results show the interaction of these clusters and their transcriptomic relationship depending on where they are localized within the regenerating intestinal tissue.

Identifying the top genes expressed by each cluster also provides essential, notwithstanding limited, information for their complete identification. To further characterize each cluster, we have used other analyses or performed additional experiments, including (1) multiple gene expression patterns, (2) enriched ontology, (3) HCR-FISH, and (4) pseudo-trajectory. Initially, we describe in depth the potential identity of these clusters based on their expression pattern and enriched ontology, and we will end with their potential interactions.

## An in-depth analysis of the various clusters and their possible relation to cell populations

### Coelomocyte populations

Coelomocytes are specialized cells found in the coelomic fluid and within organs of echinoderms. These cells have been associated with immunological roles including pathogen recognition, encapsulation, phagocytosis, debris removal, cytokine production, and secretion, among others (**Barela Hudgell et al., 2022**; **Courtney Smith et al., 2018**; **García-Arrarás et al., 2006**; **Smith et al., 1995**), In holothurians, these cells have been shown to be present at injury sites and in the mesentery and regeneration anlage (**García-Arrarás et al., 2006**). Coelomocytes can be subdivided into different populations by using morphological, physiological, and molecular characteristics (**Ramírez-Gómez et al., 2010**).

Many of the coelomocyte characteristics correlate with the top differentially expressed genes of cell populations within our data. These immune-like clusters (C6, C10, C11, and C12) have high expression of genes related to the immune system that are not shared with any other cluster. For example, C6 embodies a distinct cell population that represents a substantial number of cells (6% of all cells, 8% of the mesentery, and 4% of the cells in the anlage). These cells are the only population expressing *FBCD1* (fibrinogen C domain-containing protein 1). Cells in C6 also express other genes such as various tyrosine protein phosphate receptors, integrin alpha-8 (*ITA8*), platelet glycoprotein V (*GPV*), leucine-rich repeats, and immunoglobulin-like domains protein 2 (*LRIG2*) (*Figure 3*). The immune identity of C6 is also supported by the resulting gene set enrichment of gene ontology (gseGO) terms related to immune responses, such as *ubiquitin-dependent ERAD pathway*, *innate immune response activating cell surface receptor signaling pathway*, *respond to endoplasmic reticulum stress*, *phagocytosis*, and many more related to defense mechanisms (*Figure 3—figure supplement 2*). Similarly, C10 and C12 transcriptomic profiles suggest they correspond to immune-like populations (*Figure 3*). Specifically, C10 shows various uncharacterized genes along with *HCK* (tyrosine-protein kinase HCK), *DMBT1* (deleted in malignant brain tumors 1), *ALS* (insulin-like growth factor-binding protein complex acid labile subunit), *GPV*, and *IRF8* (interferon regulatory factor 8), *FER* (tyrosine-protein kinase Fer). The top GO-enriched terms of this cluster support its involvement in immune process, some of these being *lipase activity regulation*, *Fc receptor-mediated stimulatory signaling pathway*, *cellular pigmentation*, *B cell activation involved in immune response*, and *Fc receptor signaling pathway* (*Figure 3—figure supplement 2*). The transcripts expressed by C12 show a more complex profile, where most of the top genes are uncharacterized. Nonetheless, among the annotated genes are *BAR3* (Balbiani ring protein 3), *LYS1* (lysozyme 1), *TRPM3* (transient receptor cation channel subfamily M member 3), *PGCA* (aggrecan core protein), and *MRC1* (macrophage mannose receptor 1). The top genes of C11 include a great number of immune genes such as *MUC5A* (mucin-5AC), *FCGBP* (IgGFc-binding protein), *SSPO* (SCO-spondin), *FCN1A* (ficolin-1-A), *MUC5B* (mucin-5B), and *TIE1* (tyrosine-protein kinase receptor Tie-1). However, the top genes of this cluster also include several genes involved in neuronal activity, which is evident in the top enriched GO terms of this cluster that include *regulation of postsynaptic membrane potential*, *excitatory postsynaptic potential*, *chemical synaptic transmission*, *endoplasmic reticulum to Golgi vesicle-mediated transport*, *adult behavior*, and *regulation of neurotransmitter levels* (*Figure 3—figure supplement 2*).

To partially confirm the prediction that cells from C6, C10, C11, and C12 corresponded to coelomocyte populations, we used HCR-FISH to identify the cell types expressing the top gene in two of the clusters. We focused on the expression of *FBCD1* and *BAR3,* the genes that are the most represented by cells of C6 and C12, respectively. For each probe, in situ hybridization identified a distinct cell type in both regenerating and non-regenerating tissues (*Figure 4*, *Figure 4—figure supplements 1 and 2*). The FBC1-expressing cells showed round or oval morphologies with a central round nucleus. In some cases, short extensions could be observed. The cells were heterogeneously distributed in all tissues, including the nervous system and the body wall, and could be found associated with either epithelial tissues or with the extracellular matrix (ECM) in the normal intestine, the intestinal anlage and in the mesentery of normal or regenerating animals (*Figure 4A and C*). The BAR3-expressing cells were also distinct, isolated cells found in different tissues of the normal and regenerating animals (*Figure 4B and D*). Their numbers were not as high as those of the FBC1-labeled cells and their labeling was more punctuated within the cytoplasm. In the regenerating tissues, they were mostly associated with the mesentery. The widespread distribution of both cell types hinted at a cell function consistent with patrolling the body to detect and respond to potential threats such as injury or bacterial invasion.

## Mesenchymal cell populations

Mesenchymal cells of the intestinal anlage are yet to be well studied. They are known to be less proliferative than those in the overlying epithelium (*García-Arrarás, 1998*; *García-Arrarás et al., 2011*) and involved in ECM remodeling (*Quiñones et al., 2002*). Some of the mesenchymal cells are thought to migrate from the connective tissue of the mesentery to the connective tissue in the anlage (*Cabrera-Serrano and García-Arrarás, 2004*), while others have been shown to originate via EMT from the overlying epithelium (*García-Arrarás et al., 2011*). Some of these cells will form a mesenchymal cellular layer associated with the luminal epithelial cells as the lumen forms (*García-Arrarás et al., 2011*). As explained earlier, we propose that C2 and C7 form a separate supra-cluster, clearly

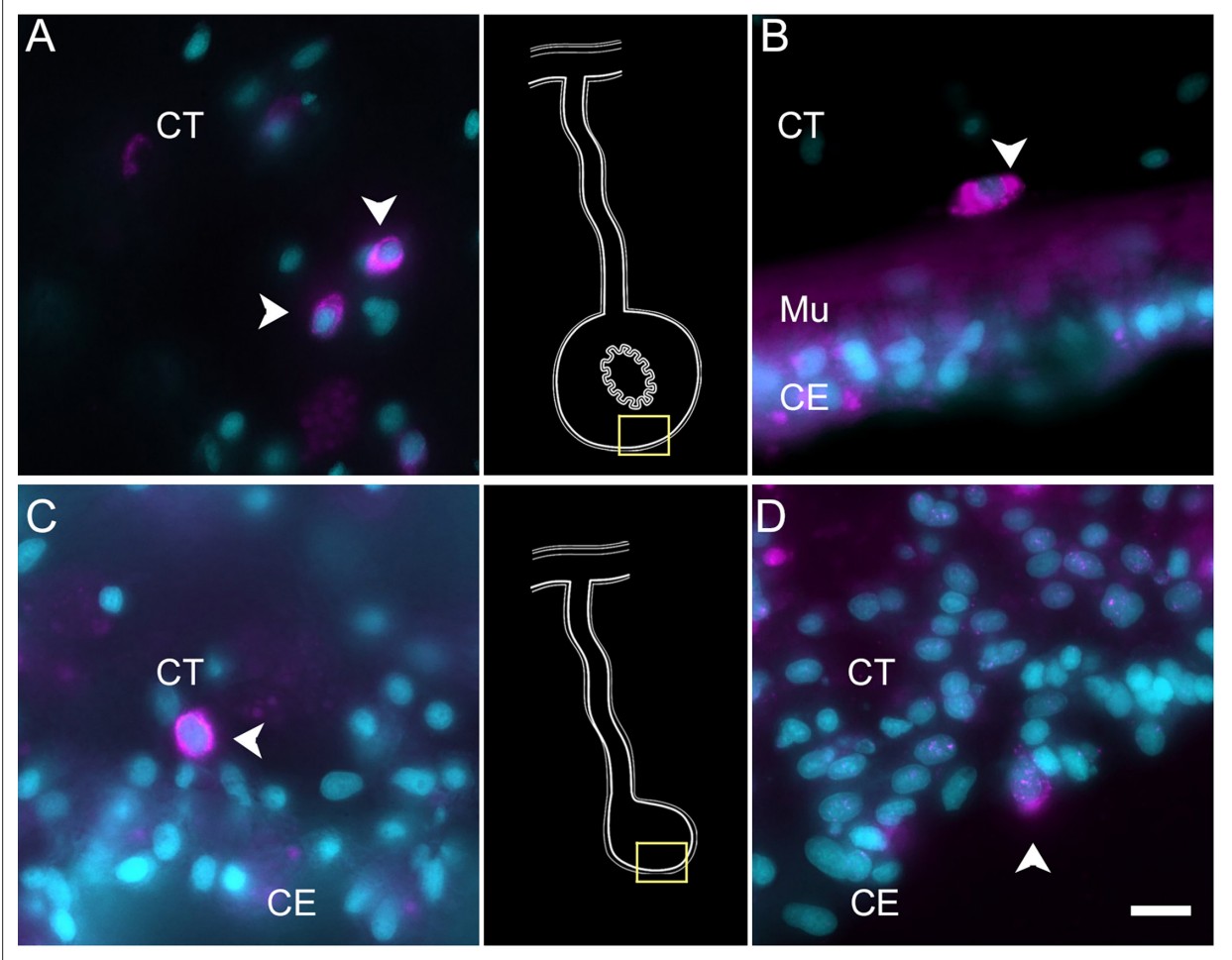

**Figure 4.** Expression profile of coelomocyte cell types. Hybridization chain reaction fluorescent in situ hybridization (HCR-FISH) for (**A, C**) FBCD1 or (**B, D**) BAR3 in holothurian (**A, B**) normal or (**C, D**) regenerating tissues. Cells (arrowheads) expressing FBCD1 mRNA in the CT layer of (**A**) normal intestine and (**C**) anlage. Cells (arrowheads) expressing BAR3 mRNA in the CT of (**B**) normal (non-eviscerated) animal and (**D**) the CE (CE) of the anlage. Insets provide the approximate localization in the CT of the normal intestine (top) and the anlage (bottom). Cyan, DAPI; magenta, HCR-FISH; CE, coelomic epithelium; CT, connective tissue; Mu, muscle. Bar = 10 μm.

The online version of this article includes the following figure supplement(s) for figure 4:

**Figure supplement 1.** Hybridization chain reaction fluorescent in situ hybridization (HCR-FISH) for Poly-A mRNA as a positive control.

**Figure supplement 2.** Hybridization chain reaction fluorescent in situ hybridization (HCR-FISH) negative control.

identified by ERG expression. These two clusters also share many marker genes, including multiple ECM genes (*Figure 3*). For instance, highest expressed gene of C2 is *TIMP3* (metalloproteinase inhibitor 3), followed by *NPNT* (nephronectin), which has been reported to be an integrin ligand during kidney development (*Sun et al., 2018*). Additionally, this cluster has overexpression of *ECM2* (extracellular matrix protein 2), *KLKB1* (plasma kallikrein), *MMP14* (matrix metalloproteinase 14), *MMP24* (matrix metalloproteinase 24), *HMCN1* (hemicentin-1), and *ITA8* (integrin alpha-8), all of which are explicitly related to the ECM component (*Bökel and Brown, 2002*; *Dolmatov and Nizhnichenko, 2023*; *Dong et al., 2006*; *Stamenkovic, 2003*; *Volkert et al., 2014*). Many of these genes are also highly represented in C7. However, here we also find *ITIH2* (inter-alpha-trypsin inhibitor heavy chain H2), *DUOX1* (dual oxidase 1), *SVEP1* (sushi, von Willebrand factor type A, EGF, and pentraxin domain-containing protein 1), *SEPP1* (selenoprotein P), and *KLH20* (Kelch-like protein 20), suggesting that cells have gained some specialization and are more advanced in their differentiation compared to those of C2. Along with this, when analyzing the gseGO, for C2 we obtain GO terms of numerous ECM processes, such as *cell adhesion mediated by integrin*, *integrin-mediated signaling pathway*, and

*regulation of cell-substrate junction assembly* (**Figure 3—figure supplement 2**). Similarly, C7 has an enrichment of *regulation of cell-matrix adhesion*, *regulation of cell-substrate junction organization*, and *substrate adhesion-dependent cell spreading. neuroblast proliferation*, among others (**Figure 3— figure supplement 2**).

To confirm our prediction that cells in C2 and C7 were those present within the mesenchyme of the connective tissue, we chose to localize the expression of *HMCN1* mRNA. The mRNA for this protein, known to code for an ECM protein (**Dolmatov and Nizhnichenko, 2023**; **Lindsay-Mosher et al., 2020**; **Welcker et al., 2021**), is present in both cell clusters. HCR-FISH showed that cells expressing the *HMCN1* mRNA were present in the normal intestine, the regenerating mesentery and the anlage (**Figure 5**). In the normal intestine and regenerating mesentery, labeled cells were present in the connective tissue (**Figure 5A and B**). In both tissues, cells were somewhat distanced from each other and had an irregular morphology with strong punctate labeling throughout the cytoplasm. A weaker labeling was observed in cells of the anlage, which were more densely packed and adjacent to the coelomic epithelium (**Figure 5C**), suggesting that they correspond to cells undergoing EMT on their way to differentiate into ECM-producing mesenchymal cells. A very similar pattern of expression was observed with HCR-FISH for *ERG* (not shown).

## Coelomic epithelia/mesothelial cell populations

The mesothelial layer of the intestine and mesentery is formed by cells (coelomic epithelium or peritoneocytes together with myocytes and neurons) in contact with the coelomic fluid. The coelomic epithelium of the regenerating tissues differs in morphology and gene expression (**Mashanov et al., 2017**; **Mashanov et al., 2005**; **Mashanov et al., 2015**; **Mashanov et al., 2012**; **Mashanov et al., 2010**) to the mesothelium that normally surrounds the organ. This coelomic epithelium, which is present mainly in the anlage and in areas of the adjacent mesentery, is made of dedifferentiated cells and is responsible for most of the cell division that takes place in the regenerating intestine (**García-Arrarás et al., 2011**). The analysis of the scRNA-seq data, in view of our knowledge of the ongoing events in the 9-dpe regenerating organ, strongly suggests that the seven clusters within the major supra-cluster represent the cells in the coelomic epithelium of the mesentery and the anlage. Here is our analysis.

### C8 represents the proliferating cells

These cells mainly found within the anlage coelomic epithelia express proliferation markers such as *PLK1* (serine/threonine-protein kinase PLK1), *SMC2* (structural maintenance of chromosomes protein 2), *PRI2* (PRIM2 – DNA primase large subunit), *PCNA* (proliferating cell nuclear antigen), and *CDK1* (cyclin-dependent kinase 1) (**Locard-Paulet et al., 2022**). In addition, this cluster has high expression of *TOP2A* (DNA topoisomerase 2-beta), a gene that has also been seen to be overexpressed in proliferating basal cells of the human gastrointestinal epithelia (**Busslinger et al., 2021**). Other marker genes related to cell mitotic activity found here include *CENPE* (centromere-associated protein E), *SMC4* (structural maintenance of chromosomes protein 4), *KI67* (proliferation marker protein Ki-67), and *CCNB3* (G2/mitotic-specific cyclin-B3) (**Figure 3**). Furthermore, this cluster expresses specific transcription factors such as *E2F3*, *MCM10* (protein MCM10 homolog), *PAF15* (PCNA-associated factor), and *BRCA1* (breast cancer type 1 susceptibility protein) that are also associated with control of cell division (**Humbert et al., 2000**; **Lõoke et al., 2017**; **Xie et al., 2014**). The gseGO terms also confirm its proliferative identity with GO terms related to *chromosome separation*, *condensation*, *mitotic cytokinesis*, and *regulation of cell cycle checkpoint* (**Figure 3—figure supplement 2**). Moreover, the dividing cell population is higher in the anlage samples (4%) than in the mesentery samples (2%), which is in accordance with what we have observed in regenerating animals, that while cell division does take place in the mesentery, more cells are proliferating in the epithelial layer of the intestinal anlage (**Bello et al., 2020**; **García-Arrarás, 1998**; **García-Arrarás et al., 2011**).

To verify the epithelial nature of this cell cluster, we performed HCR-FISH for the *KI67* mRNA. Multiple cells of the coelomic epithelium of the anlage were found to express the gene, as determined by a punctate labeling found throughout the cell body (**Figure 6A and B**). Some cells in the mesentery coelomic epithelium were also labeled but their number decreased in areas closer to the body wall. Similarly, few cells were labeled in the mesothelium of the normal intestine.

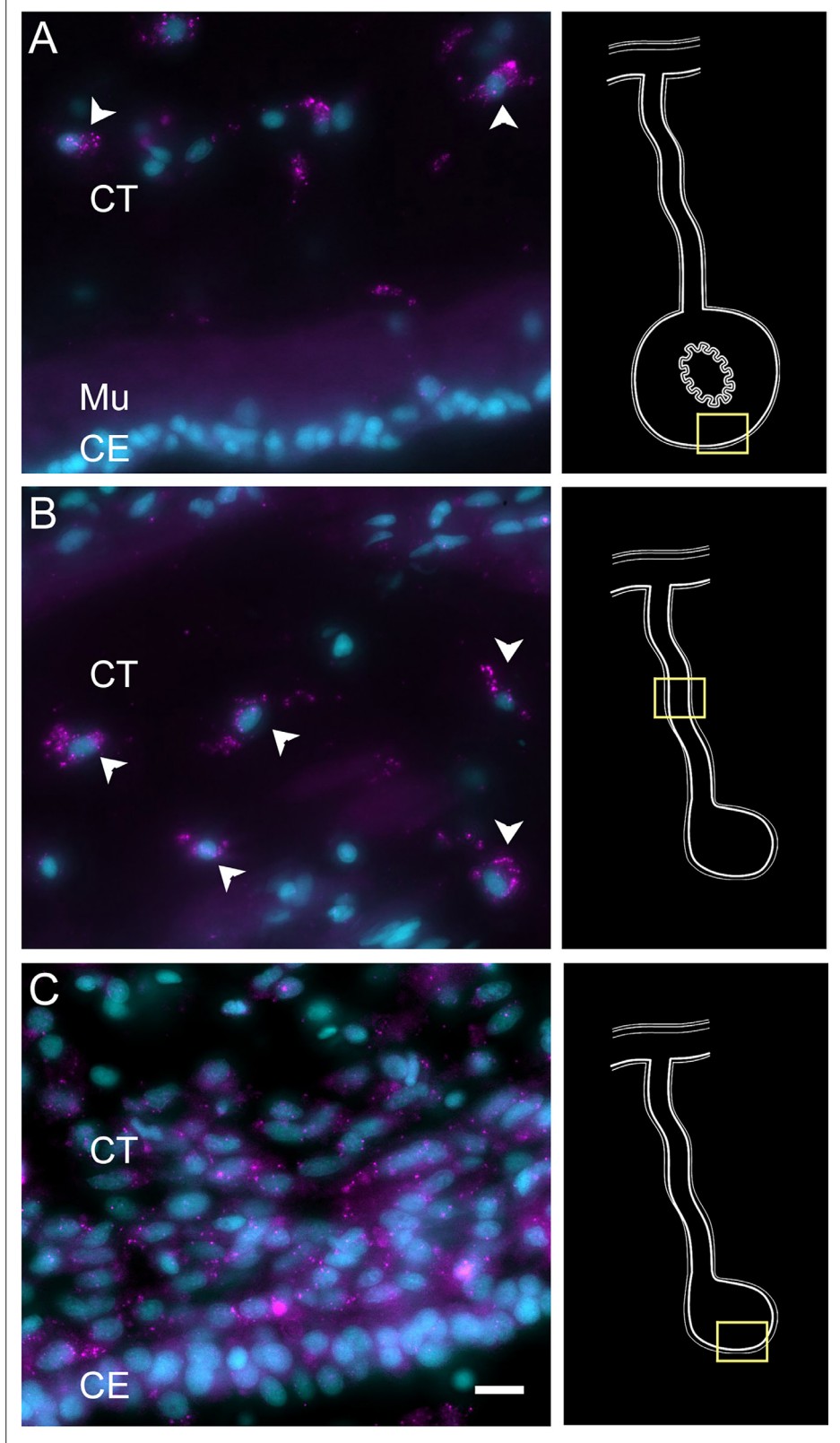

**Figure 5.** Expression profile of mesenchymal cell types. Cells (arrowheads) expressing HMCNT1 mRNA in the CT layer of (**A**) normal intestine, (**B**) regenerating mesentery, and (**C**) intestinal anlage of *H. glaberrima*. In all tissues, the cells are found within the CT layer. Notice in (**C**) that no expression is found in the cells of the CE. Insets provide the approximate localization of the cells in the adjacent photos. Cyan, DAPI; magenta, HCR-FISH; CE, coelomic epithelium; CT, connective tissue; Mu, muscle. Bar = 10 µm.

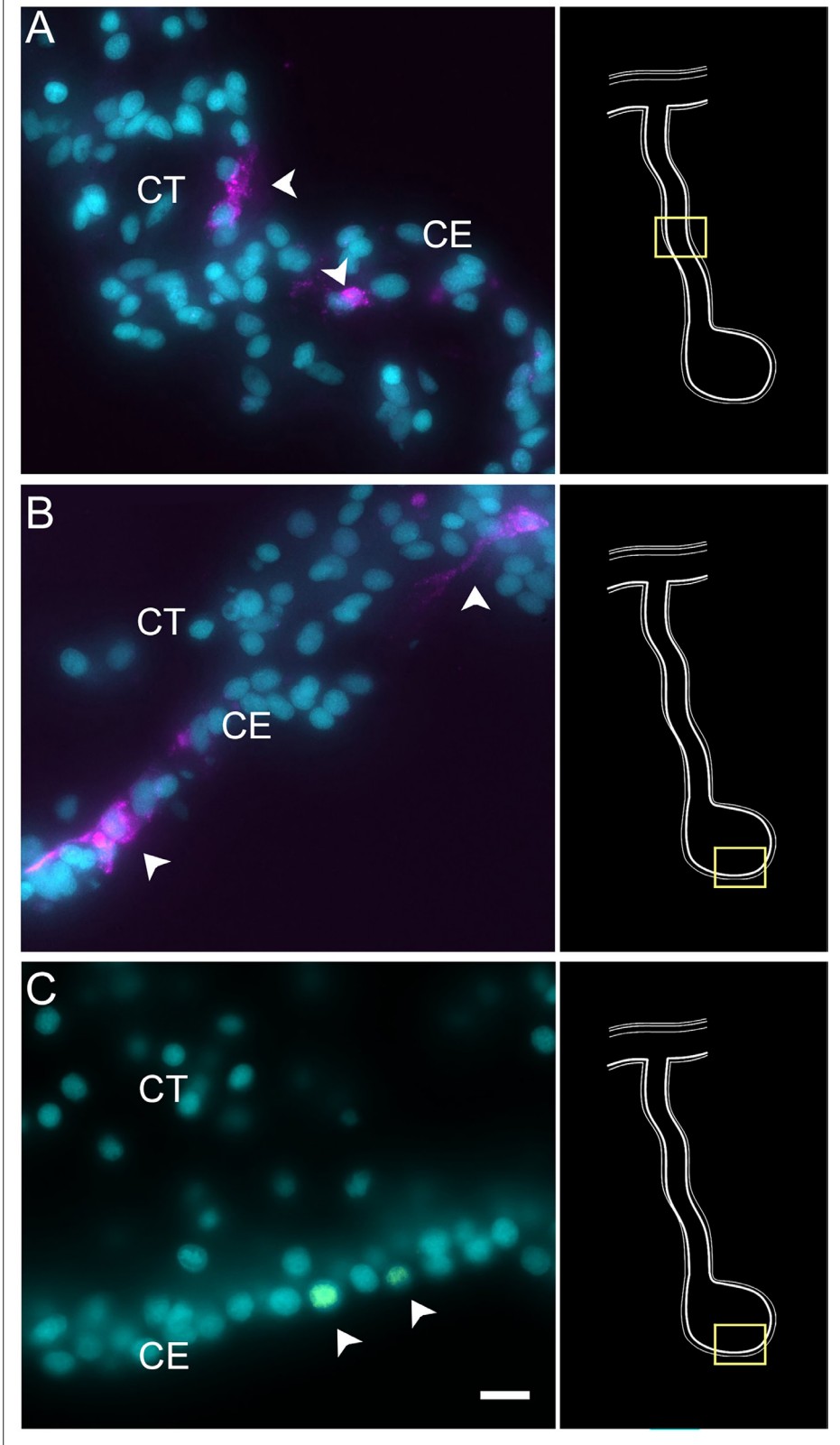

**Figure 6.** Expression profile of proliferating cells. Cells (arrowheads) expressing Ki67 mRNA in the CE layer of regenerating (**A**) mesentery and (**B**) anlage of *H. glaberrima*. Most of the labeled cells are within the CE tissue layer. (**C**) BrdU-labeled cells (arrowheads) are also mainly found in the CE layer. Insets provide the approximate localization of the cells in the adjacent photos. Cyan, DAPI; magenta, HCR-FISH; CE, coelomic epithelium; CT, connective tissue. Bar = 10 μm.

Technical aspects of the protocol do not allow for double labeling of BrdU and HCR-FISH to be performed. (The HCl step that is needed in BrdU labeling to allow the antibody access to the BrdU-labeled DNA seems to interfere with the HCR-FISH labeling.) Nonetheless, we could show that, as previously documented (*García-Arrarás, 1998*; *García-Arrarás et al., 2011*), BrdU-labeled are found in the coelomic epithelium of the anlage (*Figure 6C*). The number of BrdU cells is much smaller than that of KI67 HCR-FISH-labeled cells. This is expected since BrdU is only incorporated into the DNA in cells undergoing the S-phase while *KI67* is expressed during the whole duration of the mitotic cycle (*Kee et al., 2002*).

## C5 represents muscle precursors

Enteric muscle precursors are known to originate from the coelomic epithelium during the second week of regeneration (*Murray and García-Arrarás, 2004*). This population can be recognized in our data by the expression of muscle-specific markers present in C5, such as *TITIN*, *MYL1* (myosin light chain 1/3), *MYH7* (myosin 7), *ACTG* (actin, cytoplasmic 2), and *CNN3* (calponin-3) (*Figure 3*). Considering that this cell population is potentially a population of the coelomic epithelium actively undergoing differentiation toward muscle phenotype, they also share gene expression with other epithelial cell clusters (C0, C1, C3, C4), albeit at a lower fold-change. Similarly, this cluster shows a specific expression of transcription factors associated to muscle cells, namely *FXL16* (F-box/LRR-repeat protein 16) and *SCRT2* (transcriptional repressor scratch 2). Results of GO terms of this cluster also demonstrate enriched terms related to muscle tissue growth, such as *muscle tissue morphogenesis*, *muscle development*, *myofibril assembly*, and *sarcomere organization* (*Figure 3—figure supplement 2*). The focus on development and morphogenesis is to be expected, considering that these are still undergoing differentiation toward a muscle phenotype. HCR-FISH of *MYH7* corroborates the muscle phenotype of the cells in this cluster (*Figure 7*). In control intestinal tissues, the labeling is specific to the muscle cell layer (*Figure 7*, *Figure 7—figure supplement 1*, also see *Figure 1*). In regenerating tissues, the labeling is observed in some cells in the basal part of the coelomic epithelium (*Figure 6*), the same regions as where myoblasts or muscle cells were previously identified (*Murray and García-Arrarás, 2004*; *Figure 7*). An additional experiment was performed to double label the tissue with *MYH7* HCR-FISH and an antibody that labels the muscle fibers. The overlay of these markers showed the expression of the myosin mRNA in the normal intestine muscle cells (*Figure 7D and E*) and in cells of the regenerating tissues that have begun to form the actin-myosin contractile apparatus (*Figure 7F and G*).

## C9 represents the neuroepithelial cells

Cells in C9 represent another population of specialized cells that can be associated to cells previously described in the intestinal anlage. This small number (3%) of cells most likely corresponds to neuroepithelial cells that will eventually give rise to neurons. These cells express neuroepithelial or neuronal genes such as neurotrypsin (*NETR*), potassium gated-voltage channels, *PRD10* (*PRDM10* – PR domain zinc finger protein 10), *ELAV2* and *STA10* (STARD10 – START domain-containing protein 10) (*Figure 3*). The latter is a protein that we have characterized as being expressed by enteric neurons and nerve bundles (*Rosado-Olivieri et al., 2017*). The holothurian STA10 (STARD10) is recognized by our monoclonal antibody RN1. This antibody has been used to detect enteric neurons as they begin to differentiate in the coelomic epithelium during the second week of regeneration (*Tossas et al., 2014*). Among its associated GO terms are *positive regulation of ion transmembrane transporter activity*, *cyclic nucleotide metabolic process*, *regulation of muscle contraction*, *regulation of membrane potential*, and *positive regulation of hormone secretion* (*Figure 3—figure supplement 2*). Moreover, there is also a great representation of processes involved in development, differentiation, and growth of nerve cells, all of which together would be expected of a neuroepithelial layer. To verify the neuroepithelial nature of the cells, we performed HCR-FISH for the STA10 mRNA in the tissue and show that some cells in the regenerating mesentery and anlage coelomic epithelium express the gene (*Figure 8*). Moreover, double labeling with the HCR-FISH against STA10 and the RN1 antibody shows cells and fibers that express both the mRNA and the protein product (*Figure 8*) are forming fiber extensions typical of differentiating neurons (*Figure 8C–E*).

The characterization of these three clusters (C5, C8, C9), which represent cells undergoing differentiation or proliferation, leaves a group of four clusters (C0, C1, C3, and C4) that show some overlap

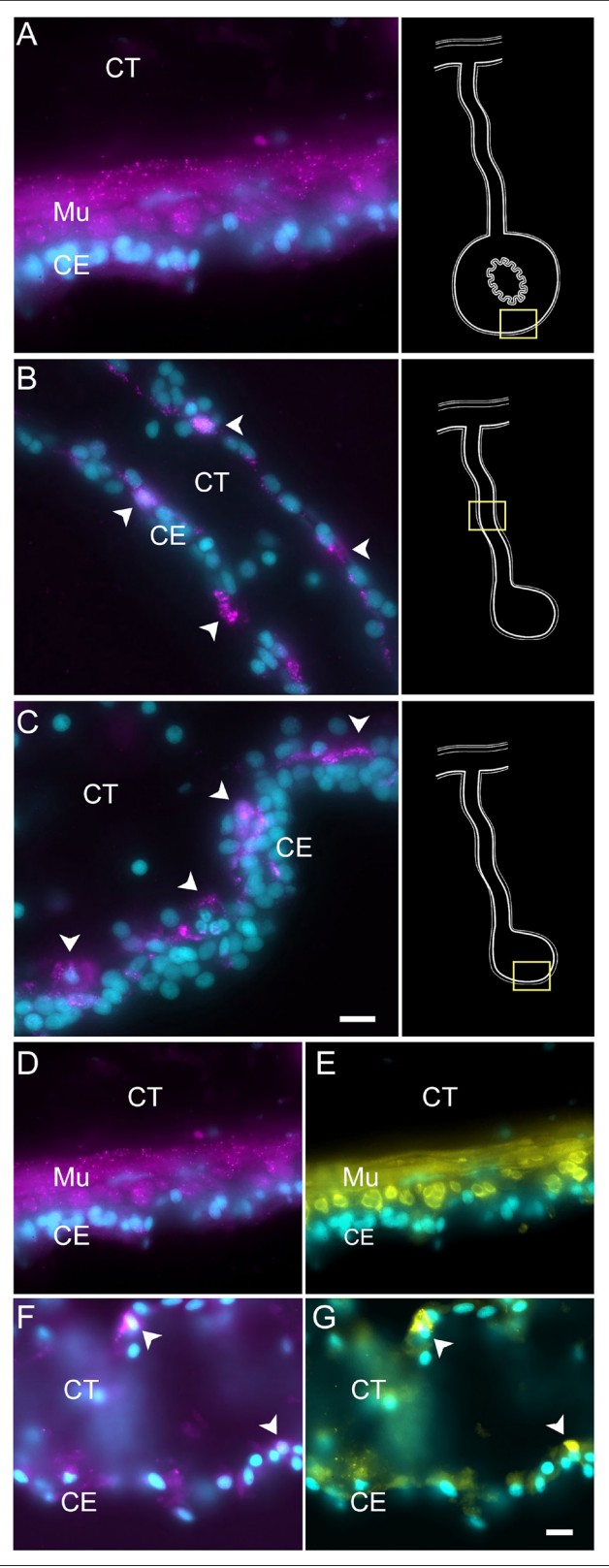

**Figure 7.** Myosin mRNA expression in regenerating and mature muscle cells. (**A–C**) Cells expressing *MYH7* mRNA (arrowheads) in the (**A**) normal intestine, (**B**) regenerating mesentery, and (**C**) basal area of the anlage CE. Arrowheads point to expression in cells of regenerating tissues. Insets provide the approximate localization of the cells in the adjacent photo. (**D–G**) Double labeling of (**D, F**) *MYH7* and (**E, G**) muscle-specific antibody (HgM2), in

*Figure 7 continued on next page*

*Figure 7 continued*

(**D, E**) normal intestine, and (**F, G**) regenerating mesentery, showing the co-expression of both markers in the same cells. Note the two cells (arrowheads in **F–G**) that express both the *MYH7* marker and the muscle-specific antibody (HgM2), representing cells initiating their differentiation toward enteric muscle cells. Cyan, DAPI; magenta, HCR-FISH; yellow, muscle antibody (HgM2); CE, coelomic epithelium; CT, connective tissue; Mu, muscle. Bars = 10 μm.

The online version of this article includes the following figure supplement(s) for figure 7:

**Figure supplement 1.** Double labeling for muscle markers in intestinal muscle cells.

in expressed genes and at the same time share some gene expression with some of the previously described clusters. Nonetheless, as seen in *Figure 3A and B*, the cells in these four clusters still have high expression and representation of specific transcripts.

## C4 represents the intestinal coelomic epithelial cells

The gene expression profile of cells in C4 sets them slightly apart from the other three clusters (C0, C1, C3). It identifies cells that are more advanced in their development toward a particular phenotype. C4 shows high expression of genes such as *KCNQ5* (potassium voltage-gated channel subfamily KQT member 5), *SC6A9* (sodium and chloride-dependent glycine transporter 1), *EFNB2* (Ephrin-B2), and *UNC5C* (Netrin receptor UNC5C) (*Figure 3*). Other than these, C4 also shows high expression of genes that are related to cell-cell interactions and ECM molecules such as *LAMA2* (laminin subunit alpha-2), *MEGF6* (multiple epidermal growth factor-like domains protein 6), *FMN1* (formin-1), and

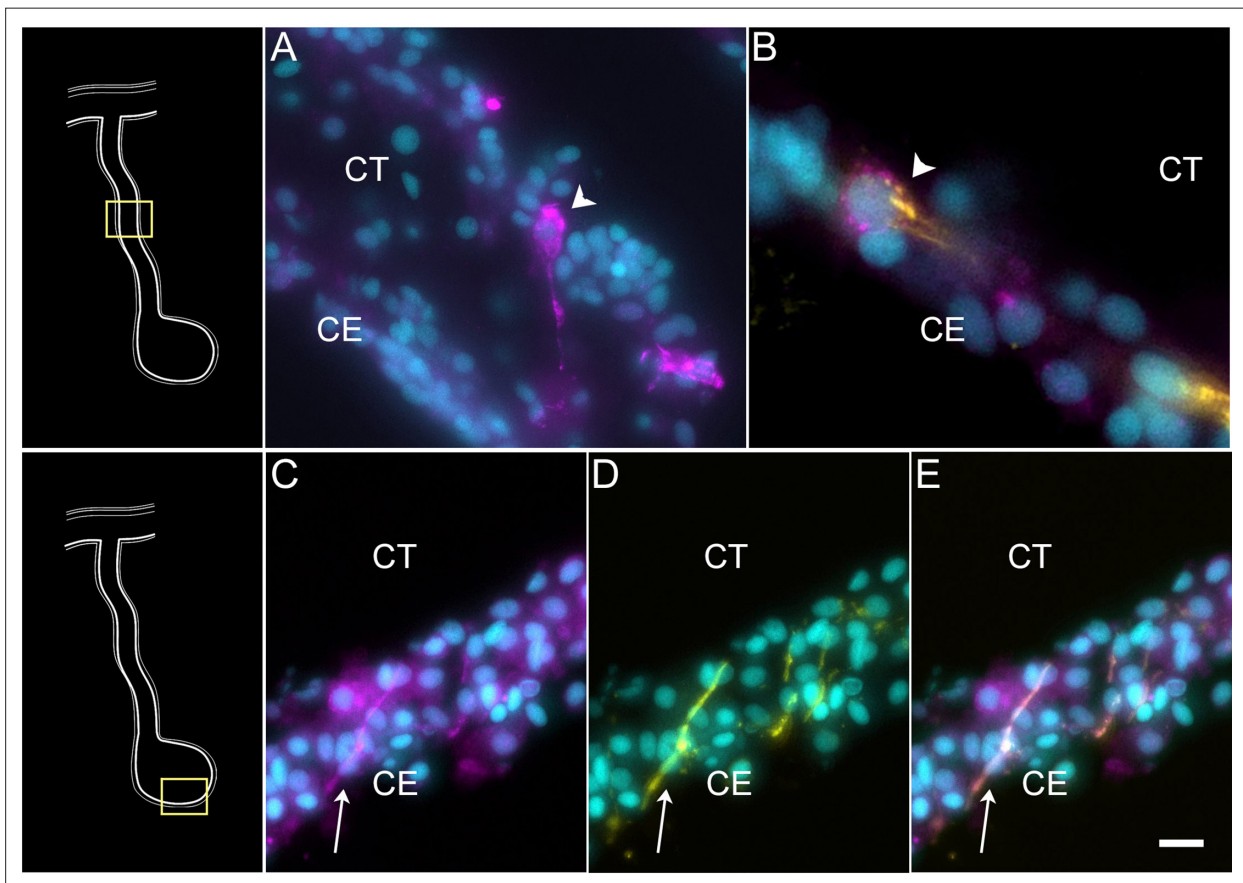

**Figure 8.** STARD-10 expression in differentiating neuroepithelial cells of the coelomic epithelium. (**A**) Cells (arrowhead) expressing *STA10* mRNA in the CE of the regenerating mesentery. (**B**) Double labeling with *STA10* hybridization chain reaction fluorescent in situ hybridization (HCR-FISH) and STARD-10 antibody (RN1) shows the presence of a cell (arrowhead) in the CE of the regenerating mesentery. (**C–E**) Fibers (arrow) in the anlage, also express both the (**C**) *STA10* mRNA and (**D**) the protein, as shown in (**E**) the corresponding overlay. Insets provide the approximate localization of the cells or fibers in the adjacent photos. Cyan, DAPI; magenta, HCR-FISH; yellow, STARD-10 antibody (RN1); CE, coelomic epithelium; CT, connective tissue. Bar: (B) = 10 μm; (A, C–E) = 5 μm.

*NPHN* (nephrin). This cluster, distinct from others, shows enrichment of GO terms related to more advanced stages of development, such as *morphogenesis of epithelium*, *sensory perception*, *regulation of calcium ion transmembrane transport*, among others (*Figure 3—figure supplement 2*). Noteworthy, cells from this cluster correspond mostly to cell from the mesentery samples (67%) rather than from the anlage (37%). Additionally, from all the mesentery cells, about 11% are part of this cluster, while only 4% of the anlage cells are represented here. HCR-FISH of *UNC5C* (Netrin Receptor), a gene differentially expressed in the cells of this cluster, provided a surprising result. The expression of this mRNA was observed as an intense labeling in cells of the normal intestine mesothelium (*Figure 9A*), In the regenerating tissue, the coelomic epithelium of the mesentery is labeled (*Figure 9B and C*) and some labeling is observed in the coelomic epithelium of the anlage (*Figure 9D*). This pattern of labeling clearly shows the cells to be part of the coelomic epithelium and strongly suggests that they correspond to cells that are in a differentiation pathway to become part of the coelomic epithelia (possibly the peritoneocytes) of the regenerated organ. This conclusion is strengthened by the pseudotime analyses presented in the following section.

## C0, C1, and C3 represent differentiation stages of coelomic epithelial cells

The three remaining clusters to be analyzed are C0, C1, and C3. These three clusters share many of their top representative genes, which are associated with developmental, regenerative, or oncogenic processes (*Bradford et al., 2009*; *Dunn et al., 2006*; *Lu et al., 2015*; *Nishimoto and Nishida, 2007*; *Oike et al., 2004*; *Zhao et al., 2008*). These include, for C0: *TRFM* (melanotransferrin), *TIMP4* (metalloproteinase inhibitor 4), and *DMBT1*; for C1: *FGF13* (fibroblast growth factor 13), *TGFB3* (transforming growth factor beta 3), *HS90A* (heat shock protein HSP 90-alpha), and *ANGL1* (angiopoietin-related protein 1) (*Figure 3*); and for C3: *SEM5B* (semaphoring-5B), *LRIG3* (leucine-rich repeats and immunoglobulin-like domains protein 3), *TUTLB* (protein turtle homolog B), and *NET1* (netrin-1). Moreover, C0 and C1 share an over-representation of GO-enriched terms that highlights biological processes related to ribosomal activity such as *cytoplasmic translation*, and *ribosome assembly*, *biogenesis*, and *assembly* (*Figure 3—figure supplement 2*). In addition to these, C3 also showed enrichment of processes involved in the development of tubular lumen-containing structures such as *mesonephric and ureteric ducts*, *differentiation and regulation of cell growth*, and *negative regulation of axogenesis* (*Figure 3—figure supplement 2*).

C1 and C3 are also closely related in their localization, being overwhelmingly associated with the anlage. These two clusters are mostly composed of cells from the anlage tissue, where we expect to see the precursor cells that will give rise to specialized cells of the organ (*Figure 1*). Precisely, C1 and C3 cells together correspond to 37% of all anlage cells compared to 14.6% of all mesentery cells. In contrast, cells from C0 correspond to 27% and 18% of the mesentery and anlage cells, respectively.

To obtain insight into these cell clusters, we performed HCR-FISH for two different mRNAs; *NET1* (Netrin), a chemotropic protein highly represented in C1 and C3, and *TRFM* (melanotransferrin) the top represented gene in C0. Both in situ hybridization experiments labeled cells in the coelomic epithelium of the regenerating intestine, supporting our contention that the large supra-cluster represents the coelomic epithelium layer (*Figure 10*). However, their spatial pattern of expression was unpredictably different. While *NET1* was highly expressed in most of the coelomic epithelial cells of the anlage, little expression was found in the regenerating mesentery or in the coelomic epithelium of the normal intestine (*Figure 10*). *TRFM*, in contrast, was highly expressed in the coelomic epithelium of the normal intestine and poorly expressed in the intestinal anlage (*Figure 10*). In the regenerating mesentery, a gradient in expression of the *TRFM* is observed, where high levels of expression were found in the coelomic epithelium close to the body wall and diminished as one approached the anlage. Thus, the HCR-FISH results show that C0, C1, and C3 correspond to cells of the coelomic epithelium, but strongly suggest that C0 differs from C1 and C3 both in their gene expression profile and in the localization where they are found, both in the regenerating and in the normal intestine.

Finally, it is essential to highlight the many signaling, or growth factors expressed by the coelomic epithelial clusters, in particular C0, C1, and C3 (*Figure 3*). These include Wnt, Hox, semaphorin, FGFs, TGF-beta, netrin, insulin-like growth factor (IGF), growth/differentiation factors (GDF), and angiopoietin-related proteins (ANGL) (e.g., *WNT9, SEM5B, FGF13, NKx3.2, TGFB3, HOX9, IGF1, GDF8, ANGL1*, and *FOXF1*) (*Figure 3*). This is important in view that the epithelium of the vertebrate blastema is characterized by its chemical modulation of the underlying mesenchyme, as will be

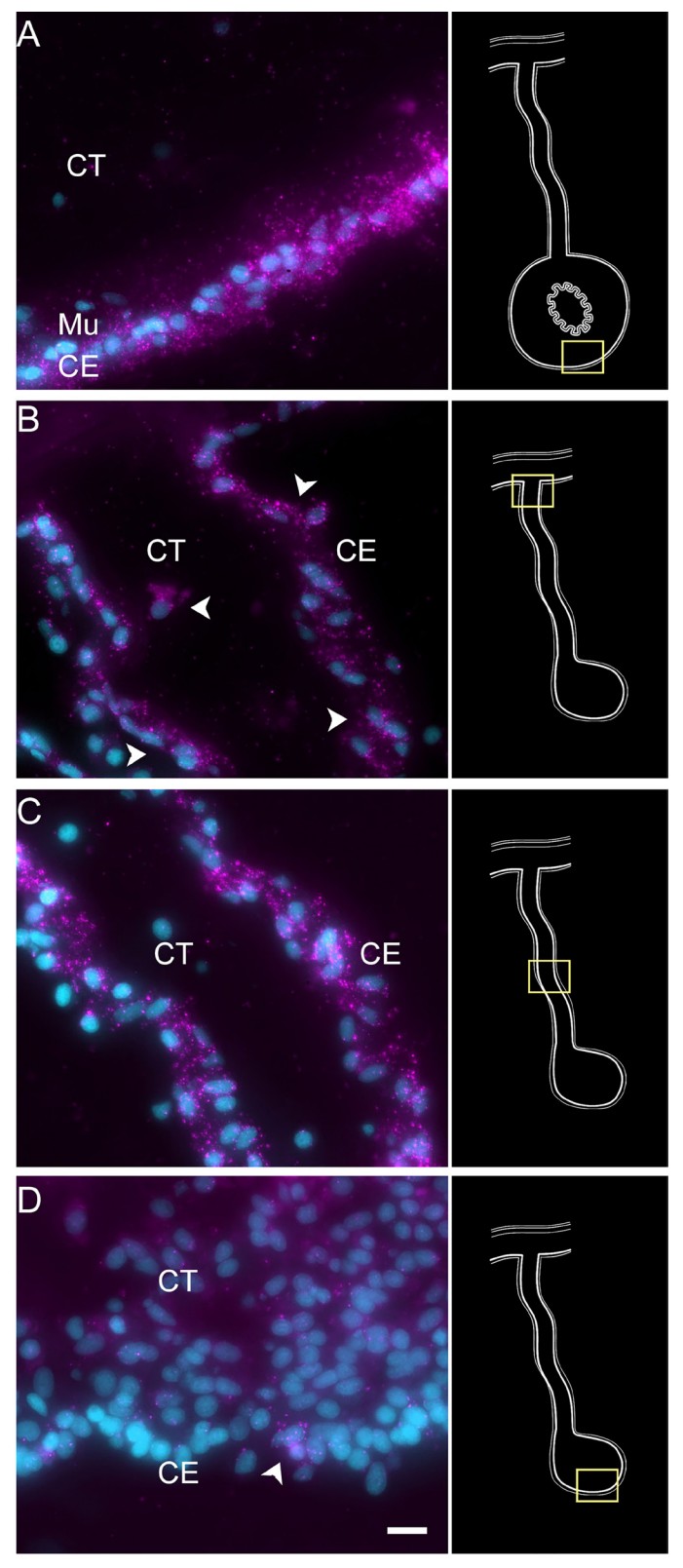

**Figure 9.** Expression profile of coelomic cell populations in the regenerating intestine anlage and mesentery. (**A**) Hybridization chain reaction fluorescent in situ hybridization (HCR-FISH) for UNC5C mRNA labeled most if not all cells in the CE of the normal intestine. Strong labeling is also observed in various regions of (**B, C**) the CE of the regenerating mesentery, while a few cells (arrowhead) are found in (**D**) the CE of the anlage (arrowhead). Insets

*Figure 9 continued on next page*

*Figure 9 continued*

provide the approximate localization of the cells in the adjacent photos. Cyan, DAPI; magenta, HCR-FISH; CE, coelomic epithelium; CT, connective tissue. Bar = 10 µm.

discussed later. Likewise, other genes that serve as markers of specific cell types or cellular stages were also identified, including *PIWL1* (piwi-like protein 1) in C1, *YAP1* in C3 and C4, *HES1* (transcription factor HES-1) in C1 and C3, and *PRRX1* (paired mesoderm homeobox protein) in C0 and C4 (*Figure 3*, *Figure 3—figure supplement 3*).

In summary, of the 13 clusters identified, our data strongly suggest that 4 of them (C9, C10, C11, and C12) correspond to coelomocytes or immune cells, 2 of them (C2 and C7) correspond to cells with a mesenchymal phenotype, and the remaining 7 to cells of the coelomic epithelia. Of these seven, C5 corresponds to differentiating muscle, C9 to differentiating neuroepithelium, C4 to differentiating coelomic epithelia, and C8 to proliferating cells. C1 and C3 represent most of the cells found in the coelomic epithelium of the anlage. In contrast, C0 represents a coelomic epithelial phenotype more closely associated with the mesentery than the intestine.

## Trajectory analysis: What populations are driving cell specification?

Among the many mysteries of the holothurian intestinal regeneration process is the identification of the precursor cells. In simple terms, what are the cells from which all nascent cells derive from? We performed a trajectory analysis of the data to provide some insights into this issue. This type of analysis is usually performed with samples at different stages or time points. However, we considered it feasible to conduct this analysis because in the 9-dpe regenerating anlage/mesentery we find cells at various stages of differentiation. These cells could provide crucial information on how the cell populations are associated with each other. To address this, we initially employed RNA velocity analysis. This method describes the temporal dynamics of gene expression based on the relative abundances of spliced and un-spliced mRNA across cell populations.

Our initial velocity analysis on all the clusters and samples (*Figure 11A and B*) provided three main results. First, the direction of arrows in our UMAP shows them flowing toward C5, C9, and C4. These arrows do not point toward any other cluster; thus, they are terminal arrows. Second, while arrows from C8 are not terminal, they are directed toward C1. This suggests that cells in C1 provide cells for the growth of the anlage via proliferation. Thus, these results support C5, C9, and C4 as terminal cell clusters that we have described as muscle, neuroepithelial, and the nascent coelomic epithelium cells, respectively. Third, velocity embedding shows shorter arrows that point from clusters 0 and 1 toward terminal populations previously described. Therefore, cells of C0 and C1 are not undergoing significant transcriptional changes. The RNA velocity results of the rest of the clusters are less interesting as they do not show directions toward any other clusters, mainly because of their individuality within the UMAP. However, it is interesting that arrows of the mesenchymal cell populations show distinct directions and lengths. Based on the results, it seems that portions of both mesenchymal clusters (C2 and C7) have gone or are undergoing more extensive differentiation changes.

We then reclustered C0, C1, and C3 cells, along with the differentiating cell populations (C4, C5, and C9) to better understand their relationship (*Figure 11C*). The velocity assessment of these newly clustered populations resulted in a similar pattern (*Figure 11D*). Here, we can see that the differentiating cells (C4, C5, and C9) have long arrows, suggesting that these cells are going through an advanced stage of transcriptional change compared to others (*Anderson et al., 2020*; *La Manno et al., 2018*). It is clear from these results that C1 future states are cells of C3 and, to some extent, C4. Moreover, it seems that C0 has a closer relationship to cells of C5 and C9 and that some cells of these clusters could potentially differentiate into cells of C4 (the coelomic epithelium cells). In this case, to complement and confirm our RNA velocity interpretation, we also performed a pseudotime analysis using Slingshot, which relies on the expression data of each cluster. This analysis showed C1 in an earlier pseudotime than C3, C4, and C0 in the resulting two lineages (*Figure 11D*). The resulting lineages differed by the terminal clusters, one containing C5 (muscle) and the other C9 (neuroepithelial). Thus, it supports what we have already visualized on the RNA velocity embeddings.

The results described so far show that C0, C1, and C4 are cells in distinct differentiation states, but we wanted to have a clearer view of the cell clusters that potentially have an essential role in the regeneration process. For this, we made another subset of cells that corresponded to C0, C1, C3, and

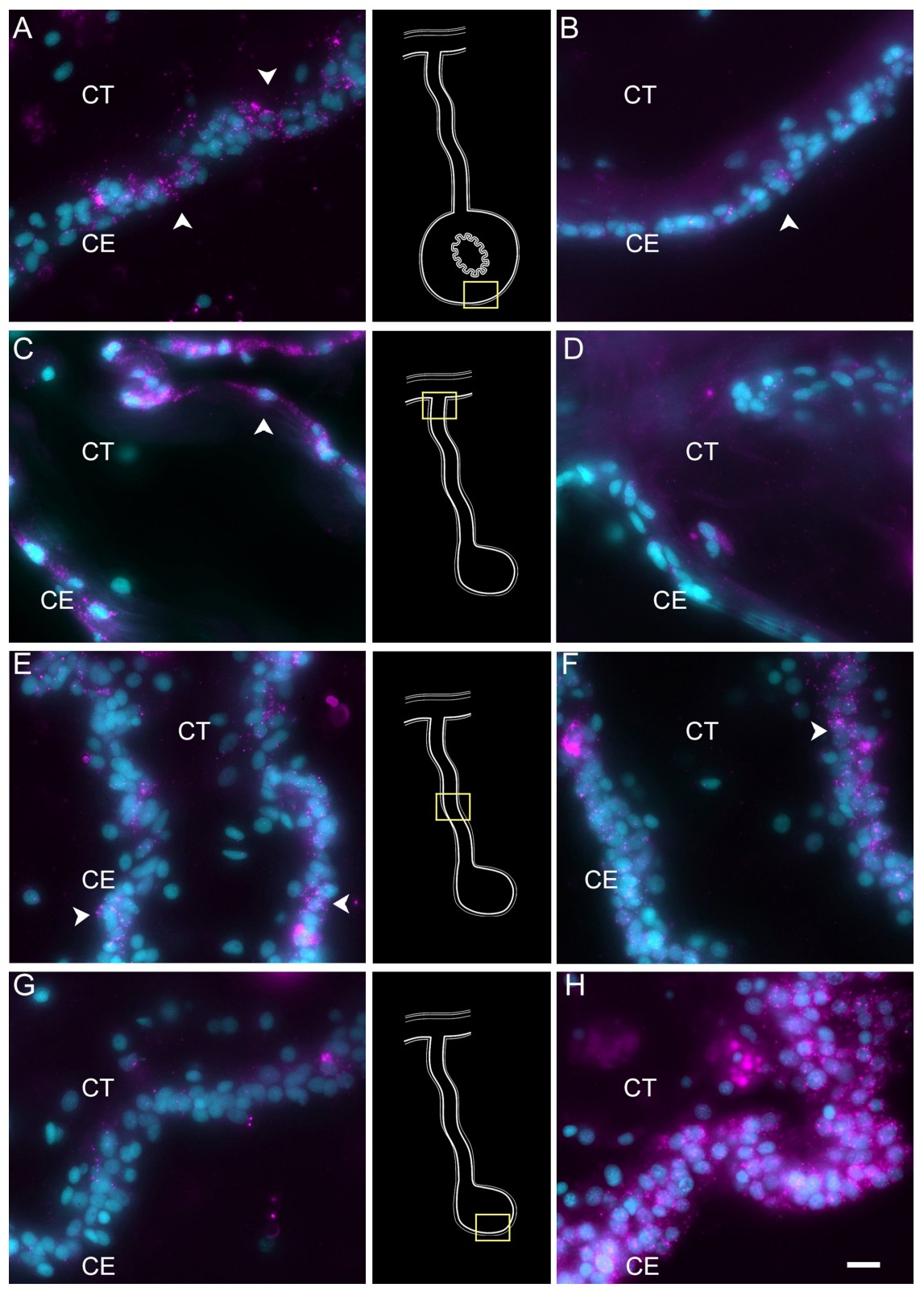

**Figure 10.** Hybridization chain reaction fluorescent in situ hybridization (HCR-FISH) for TRFM and NET1 shows the differential gene expression of the anlage versus mesentery CE. TRFM mRNA (left column) is expressed by (**A**) cells (arrowheads) of the CE in the normal intestine. In regenerating tissues, there is a high expression in (**C**) CE cells (arrowheads) close to the body wall. This expression decreases toward the anlage, with some expression in (**E**) CE cells of the mid part of the mesentery and very little if any in (**G**) the CE of the anlage. In contrast, NET1 (right column) is expressed in fewer CE

*Figure 10 continued on next page*

Figure 10 continued
cells (arrowheads) of the (**B**) normal intestine, and (**D, F**) mesentery, but its expression is extremely high in most of (**H**) the CE cells of the anlage. Insets provide the approximate localization of the cells in the adjacent photos. Cyan, DAPI; magenta, HCR-FISH; CE, coelomic epithelium; CT, connective tissue. Bar = 10 µm.

C4, but uniquely from cells of the anlage (*Figure 11F*). The rationale was that cells from the mesentery are certainly at a different state from those of the anlage and thus could interfere with the pseudo-time of cells from the anlage. The RNA velocity analysis using this subset strengthened our previous inferences. First, C1 seems to be the least dedifferentiated cell cluster, whose future state will be cells of C3 and part of the population of C4. Second, that C0 seems to be in a specialized state of differentiation that has a relationship to C4 (*Figure 11G*). This would explain the relationship of this cluster to that of the differentiating cells of C5 and C9. Interestingly, portions of C1, C0, and C4 appear to be in an advance process of differentiation based on their longer arrows compared to C3 and another portion of C4 close to C3 (*Figure 11G*). The Slingshot analysis of the anlage cells from C0, C1, C3, and C4 revealed that the pseudotime starts at a point of convergence that contains a portion of cells from C1 and C0. Yet, it further supports the cells from C1 as the least differentiated (*Figure 11H*). C1 is then followed by cells of C3, C4, and, lastly, C0, which for the most part seems to be at a more advanced differentiation state with a closer relationship to differentiating cells (*Figure 11I*).

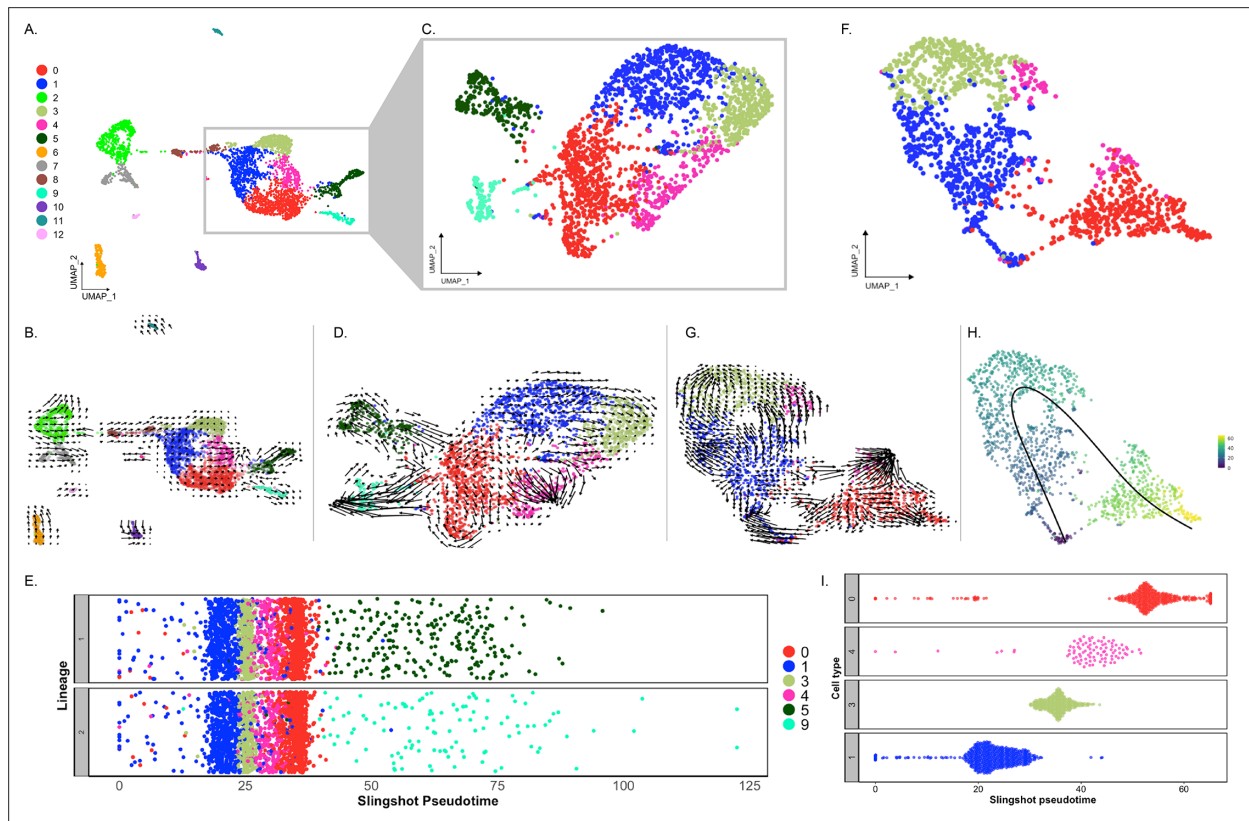

**Figure 11.** Trajectory analysis of cell populations from the regenerating intestinal tissue. (**A**) UMAP plot of all identified clusters. (**B**) RNA velocity embedded in UMAP of all main clusters. (**C**) UMAP of reclustering of cells from C0, C1, C3, C4, C5, and C9. (**D**) RNA velocity analysis results from the subset from panel (**C**). (**E**) Jitter plot of Slingshot pseudotime of cells from C0, C1, C3, C4, C5, and C9. Pseudotime resulted in two lineages, one containing C5 and the other C9. (**F**) UMAP of reclustered anlage cells corresponding to C0, C1, C3, and C4. (**G**) RNA velocity results from the cluster results from panel (**F**). (**H**) UMAP of (**F**) plot overlayed with pseudotime results of Slingshot. Color represent pseudotime values from 0 (blue) to 60 (yellow). (**I**) Jitter plot showing the Slingshot pseudotime of cells from each cell cluster.

The online version of this article includes the following figure supplement(s) for figure 11:

**Figure supplement 1.** Pseudotime analysis of all clusters within the 9-dpe intestinal regeneration dataset.

## Discussion

In this study, we employed scRNA seq and HCR-FISH techniques to examine the cellular phenotypes in regenerating intestinal tissues of the sea cucumber *H. glaberrima*. These techniques have been seldom used to characterize echinoderm cells, and the few studies available are mainly limited to embryonic stages of sea urchins (*Strongylocentrotus purpuratus, Lytechinus variegatus*) and the sea stars (*Patiria miniata*) (*Cocurullo et al., 2023*; *Foster et al., 2022*; *Meyer et al., 2023*; *Paganos et al., 2022a*; *Paganos et al., 2022b*; *Paganos et al., 2021*; *Tominaga et al., 2023*). Nonetheless, these studies provide an excellent description of the cell population and dynamics arising from major germ lines during echinoderm development. We now apply the same techniques to explore cell phenotypes involved in intestinal regeneration in holothurians. This is, to our knowledge, the first time scRNA-seq and HCR-FISH have been used in adult echinoderms to analyze the cellular and molecular basis of their amazing regenerative properties. This research integrates the extensive cellular and molecular information on intestinal regeneration in holothurians collected over the past two decades, offering a comprehensive view of the cellular phenotypes and molecular changes involved.

### Cell types and differentiation stages in the 9-dpe regenerating intestine

Our study has focused on the description of cells present in the 9-dpe regenerating intestine of the sea cucumber, where cells are known to have gone through dedifferentiation and are, at this timepoint, in the process of differentiating into specialized cells (*García-Arrarás et al., 2011*). Based on our analysis, we have identified 13 distinct populations that form part of the regenerating intestinal mesentery and anlage (*Figure 2A*). One intriguing finding was the lack of significant differences in the cell clusters between the anlage and the mesentery. However, this can be explained by two different facts. First, we have previously shown that many of the cellular processes that take place in the anlage, including cell proliferation, apoptosis, dedifferentiation, and ECM remodeling, occur in a gradient that begins at the tip of the mesentery where the anlage forms and extends significantly into the mesentery (*García-Arrarás et al., 2019*; *García-Arrarás et al., 2006*; *García-Arrarás et al., 2011*; *Mashanov et al., 2012*; *Reyes-Rivera et al., 2024*). Similarly, migrating cells move along the connective tissue of the mesentery to the anlage (*Cabrera-Serrano and García-Arrarás, 2004*). Thus, there is no clear partition of the two regions that would account for distinct cell populations associated with the regenerative stage. Second, the two cell populations that would have been found in the mesentery but not in the regenerating anlage, mature muscle and neurons, were not dissociated by our experimental protocol as to allow for their sequencing. Current experiments are being done using single-nuclei RNA sequencing to overcome this hurdle.

Among the 13 clusters we have described, the major divisions are clusters corresponding to the coelomocyte, coelomic epithelium, and mesenchyme cell types (*Figure 3*). Thus, we can now use the information that our laboratory and others have gathered to pinpoint the cellular mediators and their activity during regeneration. The coelomic epithelia, for example, is highly influential in the regeneration process as it is the site where major cellular events occur, particularly cell division, dedifferentiation, and differentiation (*García-Arrarás et al., 2011*). Moreover, the mesenchyme ECM is known to undergo remodeling during the regeneration process, which is also critical for the proper growth of the new tissue (*Quiñones et al., 2002*). However, while much of the events have been described using microscopic and histological tools, we need a more comprehensive understanding of the transcriptomic characteristics of specific cell populations involved in the regeneration process.

### The sea cucumber contains various mesenchymal and coelomocyte populations

Among the phenotypes identified are clusters of mesenchymal (C2 and C7) and coelomocyte cell populations (C6, C10, C11, and C12). The mesenchymal populations demonstrated a unique expression of *ERG* and *ETS-1*. ERG, in particular, has been related to embryonic development, differentiation, angiogenesis, and apoptosis (*Dhordain et al., 1995*; *Iwamoto et al., 2001*; *Vlaeminck-Guillem et al., 2000*; *Figure 2*). More importantly, the expression of *ERG* has been associated with mesenchymal identity in other echinoderms (*Meyer et al., 2023*; *Tominaga et al., 2023*). In these studies, *ERG* has been reported as the marker gene of mesenchymal cells of sea urchin and sea star larva by

scRNA-seq analyses, and the localization of *ERG* on embryonic precursor mesenchymal cells of the sea urchin was further confirmed by in situ analyses (*Meyer et al., 2023*). In addition, this latter group also reported these mesenchymal cells to express a GATA transcription factor (*GATA3*) and *ETS1*, which in our dataset are also being expressed only by populations within this supra-cluster (*GATA2* and *ETS1; Figure 3B*). The expression of *PRG4* (proteoglycan-4) reinforces the mesenchymal identity of this supra-cluster in the regenerating intestine (*Figure 2—figure supplement 2*), which correlates with previous studies that have identified a proteoglycan-like molecule in the mesenchyme of 7-dpe regenerating intestine (*Vázquez-Vélez et al., 2016*).

*ETS-1* has long been associated with developmental processes of mesenchymal formation in sea urchin (*Koga et al., 2010*; *Rizzo et al., 2006*). In other studies, *ERG*, a member of the ETS gene families, has been found to be necessary for controlling mesenchymal identity and differentiation (*Cox et al., 2014*; *Mochmann et al., 2014*). Further analyses of their individual marker genes suggest that these cells might be involved in EMT. For instance, C2 expresses *TIMP3*, a metalloproteinase inhibitor that aids in the ECM remodeling (*Dewing et al., 2020*), and *NPNT*, a gene reported to be related to development and cancer processes (*Magnussen et al., 2021*). These genes are also crucial for cells undergoing EMT as the cells need to detach from the other cells and the basal lamina that forms the epithelium. Comparatively, the expression profile of C7 with genes such as *PA21B* (phospholipase A2), *TMPS9* (transmembrane protease serine 9), *SEPP1*, *DMBT1*, *ITIH3*, and *SVPE1,* and its GO results suggest this mesenchymal population is undergoing different processes. Based on these contrasting expression profiles, we propose that C2 corresponds to cells that have recently undergone EMT from the coelomic epithelium and eventually differentiate into a more specialized phenotype (C7). Our pseudotime results further support this developmental transition as cells in C7 appears to be in a more advanced stage compared to those of C2 (*Figure 11—figure supplement 1*). Therefore, these two populations correspond to the first transcriptomic description of mesenchymal phenotypes reported in sea cucumber regenerating intestine.

Our dataset contains four distinct populations that we have characterized as coelomocytes. The coelomocyte populations reported in different holothuroid species ranged between 4 and 6 distinct types (*Hetzel, 1963*; *Ramírez-Gómez et al., 2010*; *Xing et al., 2008*). Studies from our laboratory previously revealed four different coelomocyte populations in *H. glaberrima*, distinguished by their morphology, histochemistry, and phagocytic activity. These were lymphocytes, phagocytes, spherulocytes, and a population named 'giant cells' (*Ramírez-Gómez et al., 2010*). The distinctive gene expression profile of each of the coelomocytes that we have identified can provide insights into the differences in their role as immune/circulating cells within the sea cucumber. For example, the C6 marker gene fibrinogen-like protein, which is part of a protein family known as FREP, makes this population of great interest. Mainly because these molecules have been vastly studied across invertebrates, and multiple immune roles have been proposed, including phagocyte recognition and encapsulation (*Hanington and Zhang, 2011*). A distinct example is that of C10, where the expression of *HCK*, *FER*, and *ITF8* markers suggests this might be a macrophage-like activity (*Chen et al., 2023*; *Dolgachev et al., 2018*; *Shuttleworth, 2018*). Specifically, *HCK*, a member of Src family kinases (SFK), has been closely related to macrophage activation and polarization (*Bhattacharjee et al., 2011*; *Poh et al., 2015*). Interestingly, recent studies in *A. japonicus* found that an Src homolog mediates the phagocytosis of *Vibrio splendidus,* which further supports C10 immune identity (*Wan et al., 2022*). Thus, to our knowledge, this would be the first report of the expression profile of distinct coelomocyte populations in an adult echinoderm species, setting up the stage for integrating these populations with the previously described ones.

## The coelomic epithelia of the intestinal anlage is composed of a heterogeneous population of cells

Our research findings align with previous microscopic descriptions of cells in the normal and regenerating mesothelium and coelomic epithelia. In our data, we can easily identify the cell population that forms the cluster exhibiting a proliferative phenotype (C8). Proliferative cells in the regenerating intestine are primarily localized within the coelomic epithelium of the anlage (*García-Arrarás, 1998*; *García-Arrarás et al., 2011*; *Reyes-Rivera et al., 2024*), and the gene expression profile documented here unequivocally identifies these as proliferative. These cell proliferation genes are specific to C8, with minimum representation in other populations. Interestingly, as mentioned before C8 expresses

at lower levels many of the genes of other coelomic epithelium populations. Nevertheless, even if we mask the top 38 proliferation genes (not shown), this cluster is maintained as an independent cluster, suggesting that its identity is conferred by a complex transcriptomic profile rather than only a few proliferation-related genes. Therefore, the identity and potential role of C8 could be further described by two distinct alternatives: (1) cells of C8 could be an intermediate state between the anlage precursor cells (discussed below) and the specialized cell populations or (2) cells of C8 are the source of the anlage precursor populations from which all other populations arise. The pseudotime data is certainly complex and challenging to interpret with our current dataset, yet the RNA velocity analysis shown in *Figure 11B* would suggests that cells of C8 transition into the anlage precursor populations, rather than being an intermediate state. This is also supported by the Slingshot pseudo-time analysis that incorporates C8 (*Figure 11—figure supplement 1*). Nevertheless, additional experiments are needed to confirm this hypothesis.

A second population of cells that can be well correlated to previously described cells is that with a muscle cell phenotype (C5). The evidence suggests that this cell population represents those cells from the coelomic epithelium that are differentiating into myocytes. This evidence includes (1) the top-expressed genes by the cells in this cluster are all muscle-associated genes; (2) the top-enriched terms are all related to muscle morphogenesis; (3) the cluster is mostly composed of cells that come from the anlage where muscle formation is known to be taking place at this stage (*Murray and García-Arrarás, 2004*); (4) differentiated muscle cells were not dissociated by the enzymatic procedure strongly suggesting that the muscle cells that we have identified in our data are those that are in a differentiation process, rather than fully differentiated cells closer to the body wall; and (5) the cells in this cluster were identified by HCR-FISH of *MYH7*. These cells are localized toward the basal region of the coelomic epithelium, the region where the differentiating myocytes are known to be present (*Murray and García-Arrarás, 2004*). Moreover, among the top expressed genes of this cluster is Troponin I (*TNNI1*), which has also been reported to be highly expressed in muscle precursor cells of the sea star embryo and immature cardiomyocytes of the chicken (*Mantri et al., 2021*; *Tominaga et al., 2023*). The high differentiation activity of these cells is also supported by our pseudotime analysis, where distinct cells within the cluster are in individual differentiation states (*Figure 11D and E*). Further analysis of these populations could allow us to understand the transcriptional changes these cells undergo to become fully specialized muscle cells.

An additional population in our dataset is the neuroepithelial population (C9). This population has *STARD10*, among its top expressed genes, known to be a phospholipid transfer protein present in a neural cell population. This population has been localized in the coelomic epithelium of the intestine, among other sites (*García-Arrarás et al., 2019*; *Rosado-Olivieri et al., 2017*). Furthermore, cells of this cluster have high expression of other genes reported to be expressed by neuronal cells of regenerative and developmental tissues, such as beta-tubulin in developing human gut and sea urchin larva, and synapsin in planaria regenerating tissue (*Cocurullo et al., 2023*; *Elmentaite et al., 2020*; *King et al., 2024*). Interestingly, an earlier study showing that beta-tubulin-positive cells arise from dedifferentiated cells of the regenerating intestinal tissue of the sea cucumber (*García-Arrarás et al., 2019*) also suggests that similar to the muscle population, these cells are neuroepithelial cells that are differentiating rather than fully specialized. Further support for this theory lies in the main contribution to C9 coming from anlage cells and their terminal differentiation state observed in the pseudotime analysis (*Figure 11D*).

The remaining four clusters of the coelomic epithelium supra-cluster are more challenging to characterize. Nonetheless, we will explore some hypotheses regarding their cellular phenotypes. The identity of C4 was put forward based on its close transcriptional and pseudotime correlation with populations of the coelomic epithelia (*Figures 3 and 11*). Additionally, in our in situ hybridization, its marker gene UNC5C as well as another gene *SC6A5* (not shown here) are mainly expressed by the coelomic epithelial cells of the normal intestine, strongly suggesting that the cells expressing this gene in the anlage are those that will become the coelomic epithelial cells of the regenerated organ (*Figure 9*).

C0 remains an intriguing cell population. On one hand, it appears to be closely related, by its gene expression, to other coelomic epithelial populations (*Figure 3*). Thus, it may represent an unknown cell population or cell stage. The in situ expression suggests that this cluster represents the cellular population of the mesentery or peritoneal coelomic epithelium. In this case, we might be

evidencing differences in the coelomic epithelium of the mesentery (exemplified by C0) from those of the coelomic epithelium of the intestine (exemplified by C4). Future experiments will be needed to address this controversy.

Lastly, C1 and C3 share many common genes and represent distinct populations but are still closely associated, regardless of the clustering parameters or the statistical assessment performed. Although we considered combining them into a single cluster, we ultimately decided against it as they likely represent different stages of cell development or plasticity in the regenerating intestine. One-to-one comparisons revealed that C1 expressed various ribosomal gene markers, while C3 expressed specific genes such as *EPHA4* (ephrin type-A receptor 4) and *LRIG3*, indicating distinct transcriptomic states. Additionally, all trajectory analyses revealed that C1 cells appear to give rise to C3 cells.

Further characterization of C1 and C3, within the anlage coelomic epithelium, suggests that these cells probably serve as cellular precursors to differentiating cells. The evidence from pseudotime analyses and gene expression of C1 and C3 strongly supports this conclusion. These cells appear to exhibit pluripotency with the potential to form muscle, neurons, coelomic epithelia, and mesenchymal cells in the regenerating intestine. Their gene expression profiles include markers from gene families associated with embryonic development such as Hox, zinc fingers, basic helix-loop-helix, and others, some of which are linked to stemness and pluripotency. For example, *HSP90A*, a molecular chaperone essential for stem cell pluripotency, and markers like *HES1* and TGFb, essential for maintaining stem cell proliferation, are present in these cell populations (*Aztekin, 2021*; *Mishra et al., 2005*). Moreover, the expression of *PIWL1* (piwi-like protein 1) and *YAP1* further supports the classification of these clusters as precursor cells of the intestinal anlage (*Figure 3—figure supplement 3*). In hydra, piwi-like molecules are exclusively localized in stem/progenitor cells (*Juliano et al., 2014*), and in mice, *YAP1* is expressed only in multipotent cells during intestinal epithelium regeneration, vital for their emergence (*Ayyaz et al., 2019*). Ultimately, trajectory analysis also supports C1 as a precursor cell population, as the dividing cells (C8) appear to give rise to C1, which then progresses toward C3. This analysis aligns with evidence of a proliferation center in the coelomic epithelia that provides precursor cells essential for the growth of the anlage.

In summary, C1 and C3 cells are probably dedifferentiated cells from the mesentery mesothelium, retaining markers common to all epithelial cells. Similar to what is known in other highly regenerative organisms, these dedifferentiated cells can proliferate and then redifferentiate into the cells of the new organ. Future experiments will determine whether they retain the memory of their previous phenotype (muscle or coelomic epithelium) or are completely pluripotent.

## The coelomic epithelium of the intestinal anlage is pluripotent

Our study reveals that the coelomic epithelium, as a tissue layer, is pluripotent. It fulfills the definition of pluripotency; a group of cells that has the ability to differentiate into many, but not all, cell types. Microscopy studies across different echinoderm species have consistently suggested that the coelomic epithelium can differentiate into various cell types. Among these are the formation of muscle cells and neurons in sea cucumbers (*Dolmatov et al., 1996*), mesenchymal cells in brittle star (*Piovani et al., 2021*), and even immune cells (coelomocytes) in sea star (*Byrne et al., 2020*; *Sharlaimova et al., 2021*). In some species, the coelomic epithelium has even been proposed to transdifferentiate into intestinal luminal cells (*Mashanov et al., 2005*). In our analysis, we can identify the cells in the coelomic epithelium that are differentiating toward muscle (C5), neurons (C9), coelomic epithelium (C4), and mesenchymal cells (C2 and C7) corroborating what has been previously described in *H. glaberrima* studies. Nonetheless, it remains unclear whether individual cells are pluripotent, thus a more definitive conclusion will be reached once lineage-tracing experiments can be done.

The coelomic epithelium of the regeneration anlage could also be the source of other cell types. This is exemplified in the brittle star *Marthasterias glacialis*, where it has been suggested that the coelomic epithelium is not only involved in arm regeneration but also serves as a source of immune cells (*Guatelli et al., 2022*). This report indicated that 'residential stem cells' in the regenerating arm of the brittle star originate from the coelomic epithelia (*Candia-Carnevali et al., 2009*). Similarly, in the sea cucumber *Holothuria forskali*, the injured mesothelial layer has been identified as the source of undifferentiated cells that differentiate into the cells of the growing organ and into phagocytic cells, now recognized as coelomocytes (*VandenSpiegel et al., 2000*).

Of particular interest, our study reveals that the mesentery coelomic epithelia population (C0) exhibits localized expression of *SAA1*, an immune response-related gene. This finding, in line with previous reports of SAA1 localization in the coelomic epithelium of the regenerating intestinal tissue (*Santiago et al., 2000*; *Figure 3—figure supplement 3*), raises the intriguing possibility that the coelomic epithelium of the sea cucumber regenerating intestine could also be the source of coelomocytes, as suggested by *Guatelli et al., 2022*. This hypothesis is further supported by the observation that in the sea star *Asteria rubens,* coelomocytes arise from its coelomic epithelium (*Vanden Bossche and Jangoux, 1976*). However, our current results, while suggestive, do not yet provide conclusive evidence to support this in the sea cucumber, at least not in the coelomic epithelia of the 9-dpe regenerating intestine.

Currently, there is no definitive evidence identifying the precursor cells that give rise to these cell populations in echinoderms. However, given our gene expression data and the interactions observed among the cell populations, we postulate that cells from C1 stand as the precursor cell population from which most of the cells in the anlage coelomic epithelium arise. This is based on their preferential localization within the anlage coelomic epithelium, their lack of any clear differentiated cell marker, and their association with the cell dividing cluster. Granted, much more experimental evidence will be needed to arrive at this conclusion. Nonetheless, it provides a focus on the most likely, and probably most intriguing, cell population. It also provides the path to explore multiple questions that arise from our results. Can C1 be subdivided into other cell populations? Are the cells in this cluster pluripotent? Do these cells originate from the same cells via dedifferentiation? These and many other questions will indeed be tested in future experiments.

## The intestinal anlage as a regeneration blastema

The intestinal anlage is a particular regenerative structure that has long been described as a blastema-like structure due to its remarkable morphological resemblance to the 'classical blastema' (*García-Arrarás, 1998*). As stated previously, a blastema is usually described as a transient structure composed of a mass of proliferating undifferentiated cells. This structure is found at the site of injury and will give rise to the regenerated organ (*Seifert and Muneoka, 2018*). The blastema cells can originate from different lineages across species. For example, in some amphibians the blastemal cells originate from dedifferentiated muscle, cartilage, fibroblast, and other tissues, while in others they include cells that originate from muscle satellite cells, a type of stem cell (*Globus et al., 1980*; *Sandoval-Guzmán et al., 2014*). In Planaria, they originate from undifferentiated stem cells known as neoblasts (*Baguñà, 2012*; *Wagner et al., 2011*). The blastema is overlayed by a wound epidermis that is formed by a re-epithelization process following injury, which develops into a specialized wound epidermis that in amphibians is known as the apical epithelial cap (AEC) (*Aztekin, 2021*). Since its first description, more than 100 years ago, the blastema has been regarded as the best indicator of regeneration, and its presence is associated with tissues or organs that are highly regenerative. In fact, it has been proposed that the presence or absence of a blastema defines the regenerative success or failure of the regeneration process. However, as more regenerative species continue to be studied, the classical definition of a blastema has been reconsidered, moving toward its functional role rather than its histological or structural characteristics (*Seifert and Muneoka, 2018*).

The blastema of salamanders and newts have long served as models to describe a blastema cellular and molecular properties (*Globus et al., 1980*; *Scimone et al., 2022*; *Tajer et al., 2023*). The sea cucumber intestinal anlage stands as a different structure in terms of the tissue compartmentalization of certain activities. Yet, when examined closely, much of the processes and signaling molecules described in the amphibian blastema also take place in the holothurian anlage, although their spatial occurrence might differ. For instance, we and others have previously described the dedifferentiation process by which muscle cells dedifferentiate and proliferate (*García-Arrarás et al., 2011*; *Dolmatov, 2021*). Thus, similar to some amphibian blastema cells, the cells of the holothurian anlage coelomic epithelium are proliferative undifferentiated cells that originated via a dedifferentiation process. Moreover, some of the genes expressed by the amphibian blastema cells or the overlying AEC, such as Wnt, Tgf-beta, and Fgf, are also known to be expressed during sea cucumber regeneration (*Auger et al., 2023*; *Zeng et al., 2023*). In amphibians, some of these factors are thought to be released by cells of the AEC, serving as a way of modulating the blastema cell activity. For example, FGF1 is expressed in the cells of the AEC while FGF receptors were localized to blastema cells (*Zenjari et al.,*

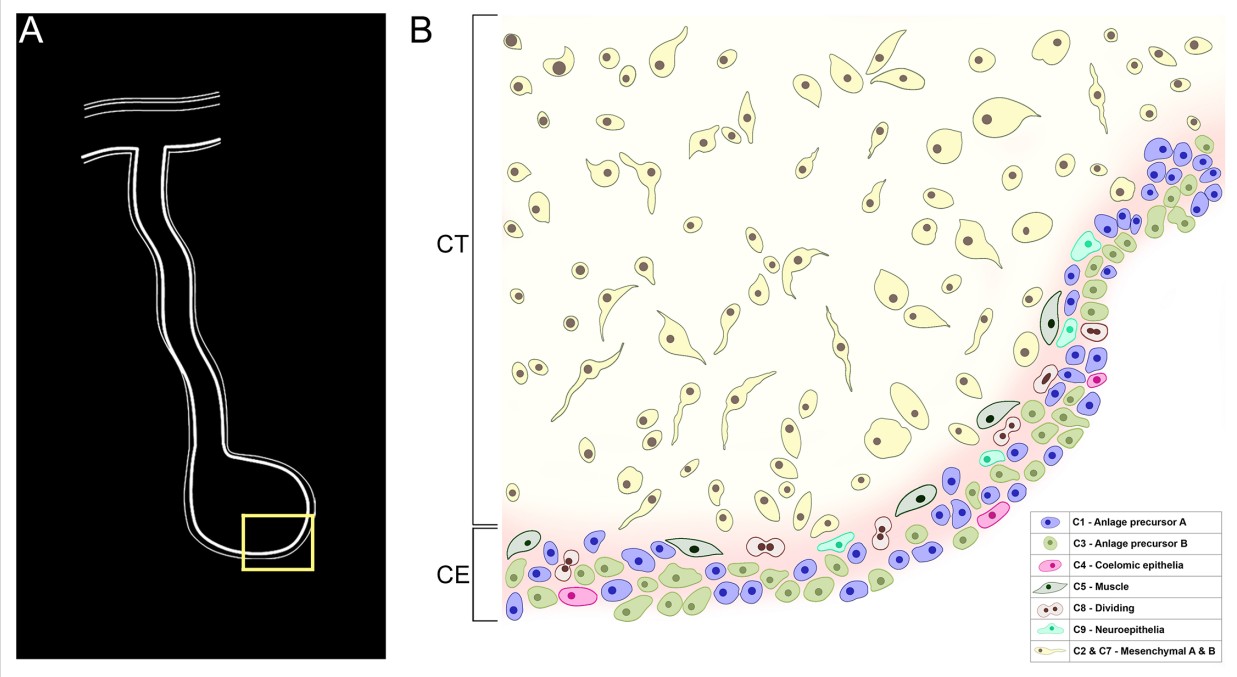

**Figure 12.** Model of cellular organization of the intestinal anlage. Various cell populations can be identified in the connective tissue (CT) and coelomic epithelium (CE) in the anlage of the sea cucumber *H. glaberrima* at 9 dpe. (**A**) A diagram showing the intestinal anlage and a portion encompassing coelomic epithelia and connective tissue that would be represented by the cell populations shown in (**B**). These populations correspond to those identified by the scRNA-seq that are described in the text. Most of the cell populations are found within the CE, some of them showing particular localizations; differentiating muscle cells are found in the basal part of the CE while differentiating coelomic epithelia are found in the apical region. The two mesenchymal populations are shown together. The colors of cells and cluster numbers (C#s) correlate with those used on the UMAP shown in *Figure 2*.

*1996*). Moreover, both FGF and Wnt signaling have been shown to be vital for the formation of the blastema (***Makanae et al., 2014***).

This expression of Fgf and Wnt and their possible functions correlate with what has been found in sea cucumbers. Not only are the same growth factors expressed in C1 and C3, but a recent study highlighted the role of *FGF4* as a modulator of cell proliferation during the intestinal regeneration of the sea cucumber *A. japonicus* (***Zeng et al., 2023***). This group demonstrated that inhibiting *FGF4* and its receptor, *FGFR2*, negatively affected cell proliferation of the mesothelial layer during intestinal regeneration. Similar results have been shown for Wnt, primarily a study by our laboratory demonstrating that Wnt pathway inhibition, either by pharmacological drugs or by RNAi, caused a reduction in cell proliferation (***Alicea-Delgado and García-Arrarás, 2021***; ***Bello et al., 2020***). Thus, in the sea cucumber coelomic epithelium of the anlage, cells are also involved in intercellular communications that modulate cellular dynamics through factors similar to those found in the AEC-blastema cell modulation.

The holothurian anlage and the amphibian blastema exhibit some differences. In the latter, there is a clear separation of the AEC and the underlying blastema cells. In amphibians, the AEC has been shown to actively participate in the formation and maintenance of the blastemal cells, particularly in their proliferation and differentiation. In contrast, in holothurians, cells with both blastema and AEC characteristics can be found within the coelomic epithelium. In fact, both *PRRX1* (paired mesoderm homeobox protein), a gene expressed by amphibian blastema cell precursors (***Gerber et al., 2018***; ***Lin et al., 2021***), and *HES1* (transcription factor HES-1), a marker gene of the amphibian AEC, are expressed by the holothurian coelomic epithelium (***Aztekin, 2021***; *Figure 3*, *Figure 3—figure supplement 3*).

Our data has allowed us to construct a model of the cell populations we identified in the 9-dpe intestinal anlage (*Figure 12*). This model presents the coelomic epithelium of the anlage as a heterogeneous layer of cells comprising of six cell populations corresponding to distinct differentiation states.

Among these are cell populations that seem to be in the process of differentiation toward muscle (C5), neuroepithelium (C9), and intestine coelomic epithelial cells (C4). We also propose the presence of undifferentiated (C0, C1, C3) and proliferating cell populations (C8) in the coelomic epithelia, which give rise to the cells in this layer. Underneath the coelomic epithelia are the mesenchymal cells (C2, C7), some of which may have originated from the coelomic epithelial layer via EMT. Moreover, the sea cucumber intestinal anlage behaves as a blastema since it uses similar mechanisms to fulfill the same role in the regeneration of the new organ. In this case, the cells of the coelomic epithelium would be those considered akin to the blastemal cells. Our results, thus, provide an alternative view of cell precursors and regenerating structures with similarities and differences to those of better-studied regenerating animal models.

## Methods

### Animals and sample collection

Two adult sea cucumbers were collected from the northeastern rocky region of Puerto Rico and were kept in aerated sea water for acclimatization prior to experiments. Evisceration of the sea cucumber was stimulated by intracoelomic injection of 0.35 M KCl as previously described (*Reyes-Rivera et al., 2024*). They were maintained in sea water aquaria for 9 days to undergo regeneration until dissection. Before the dissection, animals were anesthetized by immersion in ice-cold water for 45 minutes. Dissection was done through an initial dorsal incision which allowed exposition of the internal organs, upon which the growing anlage (or rudiment) and mesentery were dissected and separated from each other. Each tissue was kept in CMFSS on ice and treated separately during the tissue dissociation process. Thus, tissues were dissected for four scRNA-seqs; two anlage samples and two mesentery samples.

### Tissue dissociation

Mesentery and anlage tissues were digested for 15 minutes, in rocking shaker at room temperature with 1 mL of 0.05% Trypsin/0.02% EDTA solution prepared in Ca++ and Mg++ free sea water (CMFSS) (*Bello et al., 2015*). Digestions were quenched by adding 500 μL of 0.2% bovine serum albumen (BSA) in CMFSS and then centrifuged for 2 minutes at 1500 rpm. The supernatants were discarded, the pellets resuspended with 500 μL of 0.04% BSA in CMFSS, and the cells were gently separated by pipetting using glass pipettes with fire-blunted tips. The cell suspensions were filtered using a nylon cell strainer with 70 mm mesh, and aliquots were taken for cell counting. Cell viability was assessed through manual cell counting with hemocytometer to confirm that a viability higher than 90% was maintained. Sample suspensions were adjusted to 1000 cells/mL using CMFSS as required for sequencing procedures.

### Immuno- and cytochemistry

50 μL of dissociated cell sample (from the same samples that were used for scRNAseq) were placed on polysine-treated slides and fixed with 50 μL of 4% paraformaldehyde and left to dry overnight. Slides were washed in phosphate buffered saline (PBS) prior to use for immune and or cytochemistry. The methodology for the immunofluorescence techniques/preparation of slides was followed as published except for the time of PBS washes, which were of 10 minutes instead of 15 minutes (*Díaz-Balzac et al., 2007*; *Reyes-Rivera et al., 2024*). The primary and secondary antibodies used are in *Figure 1—source data 1*. Samples to be probed with fluorescent phalloidin were treated directly with Phalloidin-TRITC (1:1000; Sigma P1951) for 1 hr as described previously (*Reyes-Rivera et al., 2024*). DAPI was incorporated into the mounting medium as described previously (*Reyes-Rivera et al., 2024*). Cells were observed in a Nikon DS-Qi2 fluorescent microscope.

### Single-cell RNA sequencing and data analysis

Single-cell libraries were prepared using the 10X Genomics Chromium Next GEM Single Cell 3' Kit v3.1 and the Chromium 10X instrument, following the protocol from manufacturer (CG000204). Sequencing was carried out with Illumina NextSeq 2000 at the Sequencing and Genotyping Facility of the University of Puerto Rico Molecular Science Building.

The raw sequencing reads from all four samples (two mesentery and two anlage) were processed individually using Cell Ranger (v7.1.0) (*Zheng et al., 2017*) using the reference genome and gene models of *H. glaberrima* available at http://blastkit.hpcf.upr.edu/hglaberrima-v1. Gene models were annotated against human reference proteins from UniProt and the entire UniProt reference protein database. Gene IDs throughout the study correspond to the annotations with the human reference protein sequences from UnitProt.

Samples quality was assessed through the number of UMI, genes, and overall complexity of gene expression. The number of UMI and genes was found to be over 500 and 300, respectively (*Figure 2—figure supplement 1*). Furthermore, the distribution of genes per cells was found to be unimodal. In general aspects, samples from each animal were similar, with the total number of cells from the anlage tissue higher than that of the mesentery. General sequencing statistics after mapping reads with Cell Ranger can be found in *Figure 2—source data 1*.

Initial quality filtering was conducted in R (v4.3.2) using SoupX (v1.6.2) with default parameters to remove ambient RNA (*Young and Behjati, 2020*). Subsequent data processing was carried out using Seurat (v4.3.0.1) (*Hao et al., 2024*; *Hao et al., 2021*; *Satija et al., 2015*; *Stuart et al., 2019*). After data normalization and identifying variable features (n=2000), 22,970 anchors were identified using 20 dimensions. The datasets were integrated using the canonical correlation analysis approach of Seurat, followed by data scaling for principal component analysis (npcs = 50) and Uniform Manifold Approximation and Projection (UMAP) dimensional reduction technique (dims = 1:20).

Neighbors were then identified with 20 dimensions, leading to cluster identification at a resolution of 0.5. Various resolution parameters (0.8, 1.2, and 1.4) and dimensions (up to 35 in increments of 5) were tested before the final resolution value. The statistical analysis tool scSHC was employed to assess the probability of each cluster being unique (*Grabski et al., 2023*). Marker genes of each cluster or supra-clusters were identified using the FindMarkers() function in Seurat and mapped to the appropriate UniProt IDs. Genes highlighted throughout the study were further validated using the NCBI non-redundant reference database and EchinoBase (*Telmer et al., 2024*) via BLAST.

Differential expression data across clusters was used to perform gene set enrichment analysis of Gene Ontology (gseGO) biological processes terms using clusterProfiler (v.10.0) (*Yu et al., 2012*) with a p-value cutoff of 0.05. For this analysis, BLASTp was used to map *H. glaberrima* gene models to human reference protein sequences from NCBI (GCF_000001405.40), facilitating the assignment of ENTREZ ID to the correct human GO terms in the AnnotationDbi (v1.64.1) human database (v2.1) (*Pagès et al., 2024*).

RNA velocity loom files were generated with velocyto (v0.17), which relied on genome-masked regions obtained from RepeatModeler (v2.0.5) and RepeatMasker (v4.1.5) (*Smit et al., 2015*; *Smit and Hubley, 2015*), along with the sea cucumber gene models and CellRanger dataset. These files were further analyzed with velocyto.R (v0.6) and SeuratWrappers (v0.2.0), using the ReadVelocity and RunVelocity functions. Pseudotime analysis for the distinct data subsets was conducted using Slingshot (v2.10.0) (*Street et al., 2018*). The code of the data analysis has been made available at GitHub (copy archived at *Medina, 2024*).

## HCR-FISH

Regenerating intestines from animals eviscerated 8–9 days previously and intestines from non-eviscerated (controls) animals were collected and fixed in 4% (v/v) paraformaldehyde with phosphate-buffered saline (0.01 M PBS; 0.138 M NaCl; 0.0027 M KCl; pH7.4) overnight at 4°C. Tissues were then washed three times by PBS and treated overnight with 40% saccharose prior to cutting in the cryostat. Sections (20 mm) were prepared in a cryostat (Leica CM1850), as previously published for immunohistochemistry (*Reyes-Rivera et al., 2024*). Gene spatial expression was determined by designing 12 or 24 set split probes for each gene marker. To decrease probe unspecific binding and increase its signal to background ratio, probes were designed as described by H. Choi and colleagues (Molecular Instruments) (*Choi et al., 2018*). To ensure probe gene target specificity, all nucleotide regions selected for probe design underwent extensive validation using the newly developed *H. glaberrima* genome and transcriptome alignment tool (https://blastkit.hpcf.upr.edu/hglaberrima-v1/; *Medina-Feliciano et al., 2021*). Probes were used at a final concentration of approximately 20–25 nM.

Immediately after slide preparation, HCR-FISH v3 was carried out using a modified version of Molecular Instrument's fresh fixed frozen tissues protocol. Probe hybridization was conducted overnight at

37°C, while DNA hairpin amplification (B1-546nm or B2-546nm) was done at 25°C overnight with 3 pmol of h1 and h2, respectively. Our protocol modification involved replacing ethanol with methanol during sample permeabilization. Also, the use of proteinase K was omitted, and Tween 20 at 0.1% was added to all PBS wash buffers. Once slides were prepared, image acquisition was attained using a Nikon DS-Qi2 fluorescent microscope. Positive control probes (PolyA) were used as signal calibrators to define background from positive signal during image analysis (*Figure 4—figure supplement 2*). Negative controls with fluorescent hairpins only can be found in *Figure 7—figure supplement 1*. Each probe was studied in at least three different animals, none of which were the specimens used for the scRNA-seq.

## Acknowledgements

We thank the funding support from the National Institute of Health (NIH) under the grant number 2R15GM124595. Also, funding support was provided by the NIH Research Initiative for Scientific Enhancement (RISE) program to YMN under grant number 5R25GM061151-22. We acknowledge the High-Performance Computing Facility of the University of Puerto Rico and the Sequencing and Genotyping Facility sponsored by the University of Puerto Rico and the Institutional Development Award (IDeA) INBRE grant P20 GM103475 from the National Institute for General Medical Sciences (NIGMS), a component of the NIH and the Bioinformatics Research Core of INBRE. We acknowledge Echinobase for providing access to valuable genomic and biological data that greatly supported our research. Their resources and tools were instrumental in achieving the results presented here. We would also like to acknowledge our colleague Joseph F Ryan for his constructive feedback and computational resources.

## Additional information

### Funding

| Funder | Grant reference number | Author |
|---|---|---|
| National Institutes of Health | 5R25GM061151-22 | Yamil Miranda-Negrón |
| National Institute of General Medical Sciences | P20 GM103475 | Joshua G Medina-Feliciano<br>Griselle Valentín-Tirado<br>Kiara Luna-Martínez<br>Alejandra Beltran-Rivera<br>Yamil Miranda-Negrón<br>José E Garcia-Arraras |
| National Institutes of Health | 2R15GM124595 | Joshua G Medina-Feliciano<br>Griselle Valentín-Tirado<br>José E Garcia-Arraras |

The funders had no role in study design, data collection and interpretation, or the decision to submit the work for publication.

### Author contributions

Joshua G Medina-Feliciano, Conceptualization, Resources, Data curation, Software, Formal analysis, Validation, Investigation, Visualization, Methodology, Writing – original draft, Writing – review and editing; Griselle Valentín-Tirado, Conceptualization, Resources, Data curation, Supervision, Visualization, Methodology, Project administration, Writing – review and editing; Kiara Luna-Martínez, Investigation, Methodology; Alejandra Beltran-Rivera, Investigation; Yamil Miranda-Negrón, Methodology; José E Garcia-Arraras, Conceptualization, Resources, Data curation, Formal analysis, Supervision, Funding acquisition, Validation, Investigation, Visualization, Methodology, Writing – original draft, Project administration, Writing – review and editing

### Author ORCIDs

Joshua G Medina-Feliciano (iD) http://orcid.org/0000-0001-5678-3495
José E Garcia-Arraras (iD) https://orcid.org/0000-0001-9038-7330

## Ethics

This study involves the use of sea cucumbers (Holothuroidea), which are marine invertebrates. As such, they do not require approval by the Institutional Animal Care and Use Committees (IACUC), which generally govern vertebrate animal research. Nonetheless, all procedures involving sea cucumbers were conducted in accordance with standard scientific practices for invertebrate species, ensuring ethical treatment and minimal harm to the animals involved.

Reviewer #1 (Public review): https://doi.org/10.7554/eLife.100796.3.sa1
Reviewer #2 (Public review): https://doi.org/10.7554/eLife.100796.3.sa2
Reviewer #3 (Public review): https://doi.org/10.7554/eLife.100796.3.sa3
Author response https://doi.org/10.7554/eLife.100796.3.sa4

# Additional files

## Supplementary files

MDAR checklist

## Data availability

Data generated during this project has been made publicly available at Figshare (**Medina-Feliciano et al., 2024**) including raw and processed sequencing data. The code of the data analysis has been made available at GitHub (copy archived at **Medina, 2024**).

The following dataset was generated:

| Author(s) | Year | Dataset title | Dataset URL | Database and Identifier |
|---|---|---|---|---|
| Medina-Feliciano JG, García-Arrarás JE | 2025 | Single-cell RNA sequencing of the holothurian regenerating intestine | https://doi.org/10.6084/m9.figshare.c.7289770.v1 | figshare, 10.6084/m9.figshare.c.7289770.v1 |

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
