## [Editor Report · eLife Assessment]

This article describes a resource detailing the econstitution of *Holothuria glaberrima* gut following self-evisceration in response to a potassium chloride injection, using scRNAseq and fluorescent RNA localization in situ. It provides some new findings about organ regeneration, as well as the origins of pluripotent cells, and places these findings in the context of regeneration across species. The article’s schematic model and HCR images are a **valuable** foundation for future work. The authors provide **convincing** RNA localization images to validate their data and provide spatial context. These validation experiments are of good quality but remain challenging to connect to the complex spatial organization of complex tissues. This resource will be of interest to the field of regeneration, particularly in invertebrates, but also in comparative studies in other species, including evolutionary studies.

---

## [Referee Report · Reviewer #1 (Public review)]

Summary:

Joshua G. Medina-Feliciano et al. investigated the single-cell transcriptomic profile of holoturian regenerating intestine following evisceration, a process used to expel their viscera in response to predation. Using single-cell RNA-Sequencing and standard analysis such as "Find cluster markers", "Enrichment analysis of Gene Ontology" and "RNA velocity", they identify 13 cell clusters and their potential cell identity. Based on bioinformatic analysis they identified potentially proliferating clusters and potential trajectories of cell differentiation. This manuscript represents a useful dataset that can provide candidate cell types and cell markers for more in-depth functional analysis of the holoturian intestine regeneration.

The conclusions of this paper are supported only by bioinformatic analyses since the in vivo validation through HCR is not sufficient to support them.

Strengths:

- The Authors are providing a single-cell dataset obtained from sea cucumbers regenerating their intestines. This represents the first fundamental step to an unbiased approach to better understand this regeneration process and the cellular dynamics taking part in it.

- The Authors run all the standard analyses providing the reader with a well digested set of information about cell clusters, potential cell types, potential functions and potential cell differentiation trajectories.

Weaknesses:

- The Authors frequently report the percentage of cells with a specific feature (either labelled or expressing a certain gene or belonging to a certain cluster). This number can be misleading since that is calculated after cell dissociation and additional procedures (such as staining or sequencing and dataset cleanup) that can heavily bias the ratio between cell types. Similarly, the Authors cannot compare cell percentage between anlage and mesentery samples since that can be affected by technical aspects related to cell dissociation, tissue composition and sequencing depth.

- The Authors did not validate all the clusters.

- There is no validation of the trajectory analysis and there is no validation of the proliferating cluster with H3P or EdU co-labeling.

---

## [Referee Report · Reviewer #2 (Public review)]

Summary:

This research offers a comprehensive analysis of the regenerative process in sea cucumbers and builds upon decades of previous research. The approach involves a detailed examination using single-cell sequencing, making it a crucial reference paper while shedding new light on regeneration in this organism.

Strengths:

Detailed analysis of single-cell sequencing data and high-quality RNA localization images provide significant new insights into regeneration in sea cucumbers and, more broadly, in animals. Identifying a proliferating cluster of cells is very interesting and may open avenues to identify the cell lineage history and deeper molecular properties of the cells that regenerate the intestine.

Weaknesses:

The spatial context of the RNA localization images is challenging to interpret in this spatially complex tissue organization. Although the authors have taken care to perform RNA localization staining, it is still challenging to relate these data to their schematic model. This is only a minor weakness that will almost certainly be clarified by future work from the authors as they follow up on findings.

---

## [Referee Report · Reviewer #3 (Public review)]

Summary:

The authors have done a good job at creating a "resource" paper for the study of gut regeneration in sea cucumbers. They present a single-cell RNAseq atlas for the reconstitution of Holothuria glaberrima gut following self-evisceration in response to a potassium chloride injection. The authors provide data characterizing cellular populations and precursors of the regenerating anlage at 9 days post evisceration. As a "Tools and Resources" contribution to eLife, this work, with some revisions, could be appropriate. It will be impactful in the fields of regeneration, particularly in invertebrates, but also in comparative studies in other species, including evolutionary studies. Some of these comparative studies could extend to vertebrates and could therefore impact regenerative medicine in the future.

Strengths:

• Novel and useful information for a model organism and question for which this type of data has not yet been reported

• Single-cell gene expression data will be valuable for developing testable hypotheses in the future

• Marker genes for cell types provided to the field

• Interesting predictions about possible lineage relationships between cells during sea cucumber gut regeneration

• Authors have done a good job in the revision of making sure not to overstate the lineage claims in absence of definitive lineage-tracing experiments

• Authors have improved the figures and the overall readability of the figures and text

Specific questions:

- Is there any way to systematically compare these cells to evolutionarily-diverged cells in distant relatives to sea cucumbers? Or even on a case-by-case basis? For example, is there evidence for any of these transitory cell types to have correlate(s) in vertebrate gut regeneration?

• Authors acknowledged this would be interesting and important, but they say in the response document this is outside the scope of the current manuscript and more data would be needed to do this well.

- Line 808: The authors may make a more accurate conclusion by saying that the characteristics are similar to blastemas or behaves like a blastema rather than it is blastema. There is ambiguity about the meaning of this term in the field, but most researchers seem to currently have in mind that the "blastema" definitions includes a discrete spatial organization of cells, and here these cells are much more spread out. This could be a good opportunity for the authors to engage in this dialogue, perhaps parsing out the nuances of what a "blastema" is, what the term has traditionally referred to, and how we might consider updating this term or at least re-framing the terminology to be inclusive of functions that "blastemas" have traditionally had in the literature and how they may be dispersed over geographical space in an organism more so than the more rigid, geographically-restricted definition many researchers have in mind. However, if the authors choose to elaborate on these issues, those elaborations do belong in the discussion, and the more provisional terminology we mention here could be used throughout the paper until that element of the revised discussion is presented. We would welcome the authors to do this as a way to point the field in this direction as this is also how we view the matter. For example, some of the genes whose expression has been observed to be enriched following removal of brain tissue in axolotls (such as kazald2, Lust et al.), are also upregulated in traditional blastemas, for instance, in the limb, but we appreciate that the expression domain may not be as localized as in a limb blastema. Additionally, since there is now evidence that some aspects of progenitor cell activation even in limb regeneration extend far beyond the local site of amputation injury (Johnson et al., Payzin-Dogru et al.), there is an opportunity to connect the dots and make the claim that there could be more dispersion of "blastema function" than previously appreciated in the field. Diving a bit more into these nuances may also enable a better conceptual framework of how blastema function may evolve across vast evolutionary time and between different injury contexts in super-regenerative organisms.

• Authors addressed this comment and agree it is interesting, but given how much territory they had to cover and space limitations, they will save this type of discussion and comparative theoretical work for the future.

Overall, the manuscript is much improved.

---

## [Author Response]

The following is the authors’ response to the original reviews.

**Public Reviews**:
**Reviewer #1:**
The entire study is based on only 2 adult animals, that were used for both the single cell dataset and the HCR. Additionally, the animals were caught from the ocean preventing information about their age or their life history. This makes the n extremely small and reduces the confidence of the conclusions.

This statement is incorrect. While the scRNAseq was indeed performed in two animals (n=2), the HCR-FISH was performed in 3-5 animals (depending on the probe used). These were different animals from those used for the scRNAseq. The number of animals used has now been included in the manuscript.

All the fluorescent pictures present in this manuscript present red nuclei and green signals being not color-blind friendly. Additionally, many of the images lack sufficient quality to determine if the signal is real. Additional images of a control animal (not eviscerated) and of a negative control would help data interpretation. Finally, in many occasions a zoomed out image would help the reader to provide context and have a better understanding of where the signal is localized.

Fluorescent photos have been changed to color-blind friendly colors. Diagrams, arrows and new photos have been included as to guide readers to the signal or labeling in cells. Controls for HCR-FISH and labeling in normal intestines have been included.

**Reviewer #2:**
The spatial context of the RNA localization images is not well represented, making it difficult to understand how the schematic model was generated from the data. In addition, multiple strong statements in the conclusion should be better justified and connected to the data provided.

As explained above we have made an effort to provide a better understanding of the cellular/tissue localization of the labeled cells. Similarly, we have revised the conclusions so that the statements made are well justified.

**Reviewer #3:**
Possible theoretical advances regarding lineage trajectories of cells during sea cucumber gut regeneration, but the claims that can be made with this data alone are still predictive.

We are conscious that the results from these lineage trajectories are still predictive and have emphasized this in the text. Nonetheless, they are important part of our analyses that provide the theoretical basis for future experiments.

Better microscopy is needed for many figures to be convincing. Some minor additions to the figures will help readers understand the data more clearly.

As explained above we have made an effort to provide a better understanding of the cellular/tissue localization of the labeled cells.

**Recommendations for the authors**:**Reviewer #1 (Recommendations For The Authors)**:- Page 4, line 70-81: if the reader is not familiar with holothurian anatomy and regeneration process, this section can be complicated to fully understand. An illustration, together with clear definitions of mesothelium, coelomic epithelium, celothelium and luminal cells would help the reader.

A figure (now Figure 1) detailing the holothurian anatomy of normal and regenerating animals has been added. A figure detailing the intestinal regeneration process has also been included (S1).

- Page 5 line 92-104: this paragraph could be shortened. It would be more important to explain what the main question is the Authors would like to answer and why single cell would be the best technique to answer it, than listing previous studies that used scRNA-Seq.

The paragraph has been shortened and the focus has been shifted to the question of cellular components of regenerative tissues in holothurians.

- Page 6, line 125-127 and line 129-132: this belongs to the method section.

This information is now provided in the Materials and Methods section.

- Page 11, line 210-217: this belongs to the discussion.

This section has now been included in the Discussion.

- How many mesenteries are present in one animal?

This has now been included as part of Figure S1.

- In the methods there are no information about the quality of the dataset and the sequencing and the difference between the 2 samples coming from the 2 animals. How many cells from each sample and which is the coverage? The Authors provided this info only between mesentery and anlage but not between animals.

We have added additional information about the sequencing statistics in S4 Fig and S15 Table. Description has also been added in the methods in lines 922-926 under Single Cell RNA Sequencing and Data Analysis section.

- The result section "An in-depth analysis of the various cluster..." is particularly long and very repetitive. I would encourage to Authors to remove a lot of the details (list of genes and GO terms) that can be found in the figures and stressed only the most important elements that they will need to support their conclusions. Having full and abbreviated gene names and the long list of references makes the text difficult to read and it is challenging to identify the main point that the Authors are trying to highlight.

This section has been abbreviated.

- Figure 1: I would suggest adding a graph of holothurian anatomy before and after the evisceration to provide more context of the process we are looking at and remove 1C.

Information on the holothurian anatomy has been included in a new Fig 1 and in supplementary figure S1

- Figure 2: I would suggest removing this figure that is redundant with Figure 3 and several genes are not cluster specific. Figure 3 is doing a better job in showing similar concepts.

Figure 2 was removed and placed in the Supplement section.

- In figure 3 how were the 3 cell types defined? Was this done manually or through a bioinformatic analysis?

The cell definition was done following the analysis of the highly expressed transcripts and comparisons to what has been shown in the scientific literature.

- Figure 2O shows that one of the supra-cluster is made of C2, C7, C6 and C10. This contradicts the text page 9, line 195.

The transcript chosen for this figure gives the wrong idea that these 4 clusters are similar. We have now addressed this in the manuscript.

- Figure 4A and 4C: if these are representing a subset of Figure 3, they should be removed in one or the other. The same comment is valid also for Figures 5, 6 and 7. In general the manuscript is very redundant both in terms of Figures and text.

These are indeed subsets of Fig 3 that were added with the purpose of clarifying the findings, however, in view of the reviewer’s comment we have deleted the redundant information from all figures.

- Figure 9: since the panels are not in order, it is difficult to follow the flow of the figure. - All UMAP should have the number of the cluster on the UMAP itself instead of counting only on the color code in order to be color-blind friendly.

The figure has been modified and clusters are now identified in the UMAP by their number.

- Figure S1F seems acquired in very different conditions compared to the other images in the same figure.

Fig S1F (now S2 Fig) is an overlay of fluorescent immune-histochemistry (UV light detected) with “classical” toluidine blue labeling (visible light detected). This has now been explained in the figure legend.

- Table S7 is lacking some product numbers.

The toluidine blue product number has now been added to the table. The antibodies that lack product number correspond to antibodies generated in our lab and described in the references provided.

- The discussion is pretty long and partially redundant with the result section. I would encourage the Authors to shorten the text and shorten paragraphs that have repeating information. - It might be out of the scope of the Authors but the readers would benefit from having a manuscript that focuses more on the novel aspects discovered with the single-cell RNA-Seq and then have a review that will bring together all the literature published on this topic and integrating the single-cell data with everything that is known so far.

We have tried to shorten the discussion by eliminating redundant text.

**Reviewer #2 (Recommendations For The Authors):**
- An intriguing finding is the lack of significant difference in the cell clusters between the anlage and mesentery during regeneration. This discovery raises important questions about the regenerative process. The authors should provide a more detailed explanation of the implications of this finding. For example, does it suggest that both organs contribute equally to the regenerated tissues?

The lack of significant differences in the cell clusters between the anlage and the mesentery is somewhat surprising but can be explained by two different facts. First, we have previously shown that many of the cellular processes that take place in the anlage, including cell proliferation, apoptosis, dedifferentiation and ECM remodeling occur in a gradient that begins at the tip of the mesentery where the anlage forms and extends to various degrees into the mesentery. Similarly, migrating cells move along the connective tissue of the mesentery to the anlage. Thus, there is no clear partition of the two regions that would account for distinct cell populations associated with the regenerative stage. Second, the two cell populations that would have been found in the mesentery but not in the regenerating anlage, mature muscle and neurons, were not dissociated by our experimental protocol as to allow for their sequencing. Our current experiments are being done using single nuclei RNA sequencing to overcome this hurdle. This has now been included in the discussion.

- Proliferating cells are obviously important to the study of regeneration as it is assumed these form the regenerating tissue. The authors describe cluster 8 as the proliferative cells. Is there evidence of proliferation in other cell types or are these truly the only dividing cells? Is c8 of multiple cell types but the clustering algorithm picks up on the markers of cell division i.e. what happens if you mask cell division markers - does this cluster collapse into other cluster types? This is important as if there is only one truly proliferating cell type then this may be the origin of the regenerative tissues and is important for this study to know this.

As the reviewer highlights, we also believe this to be an important aspect to discuss. We have addressed this in the manuscript discussion with the following: “Our data suggest that there appears to be a specific population of only proliferative cells (C8) characterized by a large number of cell proliferation genes, which can be visualized by the top genes shown in Fig 3. These cell proliferation genes are specific to C8, with minimum representation in other populations. Interestingly, as mentioned before C8 expresses at lower levels many of the genes of other coelomic epithelium populations. Nevertheless, even if we mask the top 38 proliferation genes (not shown), this cluster is maintained as an independent cluster, suggesting that its identity is conferred by a complex transcriptomic profile rather than only a few proliferation-related genes. Therefore, the identity and potential role of C8 could be further described by two distinct alternatives: (1) cells of C8 could be an intermediate state between the anlage precursor cells (discussed below) and the specialized cell populations or (2) cells of C8 are the source of the anlage precursor populations from which all other populations arise. The pseudotime data is certainly complex and challenging to interpret with our current dataset, yet the RNA velocity analysis showed in Fig 11B would suggests that cells of C8 transition into the anlage precursor populations, rather than being an intermediate state. This is also supported by the Slingshot pseudotime analysis that incorporates C8 (S13 Fig).

Nevertheless, additional experiments are needed to confirm this hypothesis.”

- The schematic model presented in Fig 10 is essential for clarifying the paper's findings and will provide a crucial baseline model for future research. However, the comparison of the data shown in the HCR figures with the schematic is challenging due to the lack of spatial context in the HCR figures. The authors should find a way to provide better context in the figures, such as providing two-color in situ images to compare spatial relationships of cell types and/or including lower resolution and side-by-side fluorescent and bright field images if possible.

The figure has been modified to explain the spatial arrangement of the tissues.

The authors make several strong statements in the discussion that weren't well connected to the findings in the data. Specifically:“Regardless of which cell population is responsible for giving rise to the cells of the regenerating intestine, our study reveals that the coelomic epithelium, as a tissue layer, is pluripotent.”

This has now been expanded to better explain the statement.

738 “…we postulate that cells from C1 stand as the precursor cell population from which the rest of the cells in the coelomic epithelium arise”.

This has now been expanded to better explain the statement.

748 “differentiation: muscle, neuroepithelium, and coelomic epithelium cells. We also propose the presence of undifferentiated and proliferating cell populations in the coelomic epithelia, which give rise to the cells in this layer…”

This has now been expanded to better explain the statement.

777 “amphibians, the cells of the holothurian anlage coelomic epithelium are proliferative undifferentiated cells and originated via a dedifferentiation process…”

This has now been expanded to better explain the statement.

**Reviewer #3 (Recommendations For The Authors):**
Specific questions:- Is there any way to systematically compare these cells to evolutionarily-diverged cells in distant relatives to sea cucumbers? Or even on a case-by-case basis? For example, is there evidence for any of these transitory cell types to have correlate(s) in vertebrate gut regeneration?

This is a most interesting question but one that is perhaps a bit premature to answer due to multiple reasons. First, most of the studies in vertebrates focus on the regeneration of the luminal epithelium, a layer that we are not studying in our system since it appears later in the regeneration process. Second, there is still too little data from adult echinoderms to fully comprehend which cells are cell orthologues to vertebrates. Third, we are only analyzing one regenerative stage. It is our hope that this is just the start of a full description of what cell types/stages are found and how they function in regeneration and that this will lead us to identify the cellular orthologues among animal species.

Major revisions:- If lineage tracing is within the scope of this paper, it would provide more definitive evidence to the conclusions made about the precursor populations of the regenerating anlage.

Response: This is certainly one of the next steps, however at present, it is not possible due to technical limitations.

Minor revisions:- Line 47: "for decades" even longer! Could the authors also cite some other amphibians, such as other salamanders (newts) and larval frogs?

References have been added.

- Line 85: "specially"-could authors potentially change to "specifically"

Corrected

- Line 122: Authors should add the full words of what these abbreviations stand for in the caption for Figure 1 or in Figure 1A itself.

Corrected

- Lines 153: What conclusions are the authors trying to make from one type of tubulin presence compared to the others? It's unclear from the text.

The authors are not trying to reach any particular conclusion. They are just stating what was found using several markers, and the possibility that what might be viewed first hand as a single cell population might be more heterogenous. Although the tubulin-type information might not be relevant for the conclusions in the present manuscript, it might be important for future work on the cell types involved in the regeneration process.

- Line 226: Could the authors clarify if "WNT9" is "WNT9a". Figure 3 lists WNT9a but authors refer to WNT9 in the text.

The gene names in Fig 3 are based on the human identifiers. H. glaberrima only has one sequence of Wnt9 (Auger et al. 2023) and this sequence shares the highest similarity to human Wnt9a, thus the name in the list. We have now identified the gene as Wnt9 to avoid confusion.

- Lines 236-237: Can authors rule out that some immune cells might infiltrate the mesenchymal population?

No, this cannot be ruled out. In fact, we believe that most of the immune cells found in our scRNA-seq are indeed cells that have infiltrated the anlage and are part of the mesenchyma. This has been reported by us previously (see Garcia-Arraras et al. 2006). We have now included this in the text.

- Line 452-453: The over-representation of ribosomal genes not shown. Would it be possible to show this information in the supplementary figures?

The sentence has been modified, the data is being prepared as part of a separate publication that focuses on the ribosomal genes.

- Line 480: Could authors clarify if it's WNT9a or just WNT9?

It is indeed Wnt9. See previous response above.

- Line 500: In future experiments, it would be interesting to compare to populations at different timepoints in order see how the populations are changing or if certain precursors are activated at different times.

We fully agree with the reviewer. These are ongoing experiments or are part of new grant proposals.

- Line 567-568: Choosing 9-dpe allowed for 13 clusters, but do authors expect a different number of clusters at different timepoints as things become more terminally differentiated?

Definitely, we believe that clusters related to the different regenerative stages of cells can be found by looking at earlier or later regeneration stages of the organ. A clear example is that if the experiment is done at 14-dpe, when the lumen is forming, cells related to luminal epithelium populations will appear. It is also possible that different immune cells will be associated with the different regeneration stages.

- Line 653: References Figure 10D (not in this manuscript). Are authors referring to only 1D or 9D or an old draft figure number?

As the reviewer correctly points out, this was a mistake where the reference is to a previous draft. It has now been corrected.

- Line 701: "our study reveals that the coelomic epithelium, as a tissue layer, is pluripotent." Phrasing may be better as referring to the cell population making up the tissue layer as pluripotent/multipotent or that the cells it contains would likely be pluripotent or multipotent. Additionally, lineage tracing may be needed to definitively demonstrate this.

This has been modified.

- Line 808: The authors may make a more accurate conclusion by saying that the characteristics are similar to blastemas or behave like a blastema rather than it is blastema. There is ambiguity about the meaning of this term in the field, but most researchers seem to currently have in mind that the "blastema" definition includes a discrete spatial organization of cells, and here these cells are much more spread out. This could be a good opportunity for the authors to engage in this dialogue, perhaps parsing out the nuances of what a "blastema" is, what the term has traditionally referred to, and how we might consider updating this term or at least re-framing the terminology to be inclusive of functions that "blastemas" have traditionally had in the literature and how they may be dispersed over geographical space in an organism more so than the more rigid, geographically-restricted definition many researchers have in mind. However, if the authors choose to elaborate on these issues, those elaborations do belong in the discussion, and the more provisional terminology we mention here could be used throughout the paper until that element of the revised discussion is presented. We would welcome the authors to do this as a way to point the field in this direction as this is also how we view the matter. For example, some of the genes whose expression has been observed to be enriched following removal of brain tissue in axolotls (such as kazald2, Lust et al.), are also upregulated in traditional blastemas, for instance, in the limb, but we appreciate that the expression domain may not be as localized as in a limb blastema. Additionally, since there is now evidence that some aspects of progenitor cell activation even in limb regeneration extend far beyond the local site of amputation injury (Johnson et al., Payzin-Dogru et al.), there is an opportunity to connect the dots and make the claim that there could be more dispersion of "blastema function" than previously appreciated in the field. Diving a bit more into these nuances may also enable better conceptual framework of how blastema function may evolve across vast evolutionary time and between different injury contexts in super-regenerative organisms.

We have followed the reviewer’s suggestion and stated that the holothurian anlage behaves as a blastema. Though we would love to elaborate on the blastema topic, as suggested by the reviewer, we believe that it would extend the discussion too much and that the topic might be better served in a different publication.

- In the discussion, it would be important not to leave the reader with the impression that all amphibian blastema cells originate via dedifferentiation. This is not the case. For example, in axolotls (Sandoval-Guzman et al.) and in larval/juvenile newts, muscle progenitors within the blastema structure have been shown to originate from muscle satellite cells, a kind of stem cell, in stump tissues (while adult newts use dedifferentiation of myofibers to generate muscle progenitors in the blastema). Most cell lineages simply have not been evaluated in the level of detail that would be required to definitively conclude one way or the other, and the door is open for a more substantial contribution from stem cell populations than previously appreciated especially because new tools exist to detect and study them. Providing the reader with a more nuanced view of this situation will not negatively impact the findings in this paper, but it will show that there is biological complexity still waiting to be discovered and that we don't have all the answers at this point.

This has now been corrected.

Figures: Overall, the figures need minor work.- Figure 1A: Can the authors draw a smaller, full-body cartoon and feature the current high-mag cartoon as an inset to that? Can they label the axes and make it clear how the geometry works here?

Fig 1 has been re-done and now is split into Fig 1 and Fig 2.

- Figure 1B: Can the authors label the UMAP with cluster identities on the map itself? This will make it easier to identify each cluster (especially to make sure cluster 11 is easier to find).

This has been corrected.

- Figure 2: Could the authors put boxes/clearly distinguish panel labels around each cluster (AO), so that there are clear boundaries?

Fig 2 has been moved to Supplement, following another reviewer recommendation.

- "Gene identifiers starting with "g" correspond to uncharacterized gene models of H. glaberrima." - The sentence is from another figure caption but this figure would benefit from having this sentence in the figure caption as well.

This has been added to other figures as suggested.

- Figure 3A: Can the authors potentially bold, highlight, or underline genes you discuss in text, so it's easier for the reader to reference?

This has been added as suggested.

- Figure 3C: Can the authors please label the cell types directly on the UMAP here as well?

The changes were made following the reviewer’s recommendation.

- Figure 4D-E: There's not much context here to determine if this HCR-FISH validation can tell us anything about these cells besides some of them appear to be there. Do authors expect the coelomocyte morphology to look different in regenerating/injured tissue versus normal animals? Can the authors provide some double in situs, as well as some lower-magnification views showing where the higher-magnification insets are located? Is there any spatial pattern to where these cells are found? Counter stains would be helpful.- Figure 6C: If clusters C5, C8, C9 are part of the coelomic epithelium, then authors could show a smaller diagram above with blue and grey to show types and then show clusters separately to help get their point across better.- Figure 6G: This image appears to have high background- would it be possible for authors to repeat phalloidin stain or reimage with a lower exposure/gain. Additionally, imaging with Zstacks would help to obtain maximum intensity projections. It would greatly aid the reader if each image was labeled with HCR probes/antibodies that have been applied to the sample.- Figure 7E: The cells appear to be out of focus and have high background. Additionally, they are lacking the speckled appearance expected to be seen with HCR-FISH. Would it be possible for authors to collect another image utilizing z-stacks?

HCR-FISH figures identifying the gene expression characteristic of cell clusters have been modified following the reviewer’s concerns. The changes include:

(1) Additional clusters have been verified with probes to gene identifiers. These include clusters 8, 9 and 12.

(2) Redundant information has been removed.

(3) Colors have been changed to make figures friendlier to color-impaired readers.

(4) Spatial context has been added or identified.

(5) In some cases, improved photos have been added

(6) Better labels have been included

(7) When necessary individual photos used for the overlay have been included.

- Figure 9A: Could authors add cluster labels onto UMAP directly?

This change was made to Fig 2A. UMAP in Fig 9A is the same and used just as reference of the subset.

- Figure 10: It could be useful if authors put a small map of the sea cucumber like in other images so that readers know where in the anlage this zoomed in model represents.

Added as suggested by the reviewer.

- Supplementary figure 1F: Could authors add an arrow to the dark cell that's being pointed out?

Changed made as suggested by the reviewer.

- Supplementary figure 1: Could authors label clearly what color is labeled with what marker?

Changed made as suggested by the reviewer.